# Cell-intrinsic Aryl Hydrocarbon Receptor signalling is required for the resolution of injury-induced colonic stem cells

Kathleen Shah[1,5], Muralidhara Rao Maradana[1,5], M. Joaquina Delàs [1], Amina Metidji[2], Frederike Graelmann [3], Miriam Llorian [1], Probir Chakravarty[1], Ying Li[1], Mauro Tolaini[1], Michael Shapiro[1], Gavin Kelly [1], Chris Cheshire[1], Deendyal Bhurta[4], Sandip B. Bharate[4] & Brigitta Stockinger [1✉]

The aryl hydrocarbon receptor (AHR) is an environmental sensor that integrates microbial and dietary cues to influence physiological processes within the intestinal microenvironment, protecting against colitis and colitis-associated colorectal cancer development. Rapid tissue regeneration upon injury is important for the reinstatement of barrier integrity and its dysregulation promotes malignant transformation. Here we show that AHR is important for the termination of the regenerative response and the reacquisition of mature epithelial cell identity post injury in vivo and in organoid cultures in vitro. Using an integrative multi-omics approach in colon organoids, we show that AHR is required for timely termination of the regenerative response through direct regulation of transcription factors involved in epithelial cell differentiation as well as restriction of chromatin accessibility to regeneration-associated *Yap/Tead* transcriptional targets. Safeguarding a regulated regenerative response places AHR at a pivotal position in the delicate balance between controlled regeneration and malignant transformation.

[1] The Francis Crick Institute, London, UK. [2] Department of Oncology, St Jude Children's Hospital, Memphis, TN, USA. [3] Immunology and Environment, Life & Medical Sciences (LIMES) Institute, University of Bonn, Bonn, Germany. [4] Natural Products & Medicinal Chemistry Division, CSIR - Indian Institute of Integrative Medicine, Canal Road, Jammu 180001, India. [5] These authors contributed equally: Kathleen Shah, Muralidhara Rao Maradana. ✉email: Brigitta.stockinger@crick.ac.uk

The epithelium of the gastrointestinal tract has a remarkable ability for self-renewal, which results in replenishment of epithelial cells from intestinal stem cells (ISC) every 3–4 days. Recent studies have shown that even mature epithelial cells possess extensive plasticity which allows them to efficiently regenerate following intestinal injury[1–6]. Following DSS-induced damage in the colon, this process involves dedifferentiation and reprogramming of committed epithelial cells back into a fetal-like state, orchestrated by changes to the extracellular matrix (ECM) and activation of the mechano-sensing transcriptional activators *Yap* and *Taz*[1]. Dysregulation of the pathways involved in tissue repair and differentiation underlies inflammatory disorders of the gastrointestinal tract and the susceptibility to malignant transformation[2,3].

The increase in inflammatory disorders of the gastrointestinal tract over the last 50 years suggests that environmental factors have a major role in triggering or exacerbating such diseases[7] but on the other hand environmental factors also shape physiological processes in the gut. Our focus as a molecular entry point for environmental factors is the aryl hydrocarbon receptor (AHR). AHR is a ligand activated transcription factor that upon ligand binding translocates from the cytoplasm to the nucleus where it dimerises with its co-factor ARNT to enable binding to DNA. Initially studied as the conduit for toxic effects of man-made pollutants such as dioxin, it has become clear in the last 10 years that AHR responds to endogenous ligands derived from dietary and microbiota metabolites and affects numerous physiological processes in the body[8]. Deficiency in AHR or alterations in the AHR pathway are associated with increased inflammatory responses particularly in the gut environment[9] and AHR deficient mice are highly susceptible to gut infections[10] or epithelial damage inflicted by dextran sulfate sodium (DSS), a mouse colitis model[11,12]. So far, most of the studies on AHR functions in the gut have focussed on immune cell types, where it is involved in the maintenance of innate lymphoid cells (ILC3) or intraepithelial lymphocytes[13–15] as well as in expression of the cytokine IL-22 which plays an important role in barrier defence[10,16–18]. However, using mice with epithelial-specific AHR deletion, we have previously shown that AHR also plays a cell intrinsic role in intestinal epithelial cells (IEC). Absence of AHR in IEC dysregulates intestinal stem cell differentiation and impacts transcriptional activation of *Znrf3* and *Rnf43* which are negative regulators of the Wnt pathway, causing exaggerated activation of Wnt signalling in intestinal stem cells in a DSS/azoxymethane (AOM) colon cancer model[19].

In the present study, we investigated AHR dependent events following acute injury to the epithelial barrier and show that AHR is a key determinant for the resolution of the regenerative response. Tissue damage caused by dextran sulphate sodium (DSS) colitis model leads to transient reprogramming of the intestinal epithelium to a 'fetal-like' state, characterised by the expression of stem cells antigen-1 (Sca-1), in a YAP1-dependent manner in mice. We provide evidence for AHR involvement in the resolution of this program in an epithelial-intrinsic manner using Vil-cre AHR^fl/fl mice. Loss of AHR caused failure to terminate the regeneration program with persistent Sca-1 expression and impaired reacquisition of a mature epithelial cell identity. In order to decipher the mechanisms underlying AHR mediated effects in epithelial regeneration, we generated colon organoid cultures which recapitulate aspects of epithelial injury and regeneration[20,21]. Integration of RNAseq, ChIPseq and ATACseq datasets comparing wildtype and AHR deficient organoids were used to gain insight into regeneration and differentiation pathways affected by loss of AHR. We show that AHR is required to restrict chromatin accessibility to regeneration-associated genes such as *Yap1/Tead* and associated factors. Furthermore, AHR

functions as transcriptional repressor for proto-oncogenes such as *Sox9* and *Myc*, thereby terminating the repair response and allowing the switch to differentiation. Conversely, AHR acted as transcriptional activator for key factors in the specification of intestinal epithelial cell fate, such as *Cdx2*, facilitating the efficient and timely differentiation of cells post regeneration. Without AHR, this process was defective, leading to a prolonged regenerative program at the expense of differentiation, a scenario that may underlie the increased risk of colorectal cancer initiation in AHR deficient mice[19,22]. Thus, AHR is a key factor driving the re-establishment of intestinal identity upon exit from the regenerative program. Physiological AHR activation by endogenous ligands such as dietary or microbiota metabolites safeguards efficient barrier reconstitution following injury, emphasising that important regenerative processes in the gut are influenced by environmental signals that are to some extent under our control.

## Results

**AHR drives termination of colonic epithelial regeneration.** AHR is ubiquitously expressed in colon epithelium[10]. To clarify the role of AHR in epithelial cell maintenance under steady-state conditions, we compared the number of mature epithelial subsets and proliferating cells between Vil-cre AHR^fl/fl mice and their wildtype littermate controls. Epithelial-intrinsic loss of AHR at steady state did not impact the differentiation of colonic epithelial subsets, nor result in hyperproliferation (Supplementary Fig. 1a–d), suggesting that AHR regulation of this process is dispensable, at least under the steady state conditions within our animal facility. The protective roles of AHR in epithelial barrier function have largely been demonstrated in the context of barrier perturbation by infection or following injury with DSS[10,12,16,19,22], indicating that AHR may have a prominent role in the reacquisition of colonic epithelial homeostasis post-injury; a process whose dysregulation could render epithelial cells susceptible to malignant transformation[2,3]. To investigate the epithelial-intrinsic role for AHR during tissue regeneration, we characterized the response of Vil-cre AHR^fl/fl and littermate-controls to an acute model of colonic epithelial injury (2% DSS) for 5 days.

Expression of the regenerative marker Sca-1 was evident in injured epithelial cells as early as day 4 post-DSS administration (Fig. 1a) in line with previous reports[1]. Furthermore, epithelial cells within these regenerative foci exhibited marked tissue hyperplasia, alterations to crypt morphology and mucosal thickening characteristic of an epithelial wound repair response (Fig. 1b). This was accompanied by a loss of differentiated cells such as mucus secreting goblet cells (Supplementary Fig. 1f). In both WT and Vil-cre AHR^fl/fl mice, peak epithelial Sca-1 expression was observed at day 12 post-DSS administration, with epithelial cells in Vil-cre AHR^fl/fl having significantly higher Sca-1 expression compared to WT littermates (Fig. 1c). By day 21, regenerative foci were resolved in WT control mice but persisted in Vil-cre AHR^fl/fl mice even at D30 post-injury. This was accompanied by the persistent reduction of markers expressed by mature epithelial cells such as Muc2 and Krt20 (Fig. 1d), indicating that an epithelial-intrinsic loss in AHR activity results in an impaired termination of the regenerative response albeit this did not differentially affect disease parameters such as weight or colon length (Supplementary Fig. 1e).

AHR functions are ligand dependent and therefore any reduction in ligands will likely impact functional outcomes. This is illustrated by a mouse model in which Cyp1a1 expression in intestinal epithelial cells is constitutive due to its placement under control of the ubiquitously active Rosa26 promoter (Villin-Cre R26^LSL-Cyp1a1. This leads to rapid metabolic clearance of AHR ligands, compromising AHR activation and thereby phenocopying

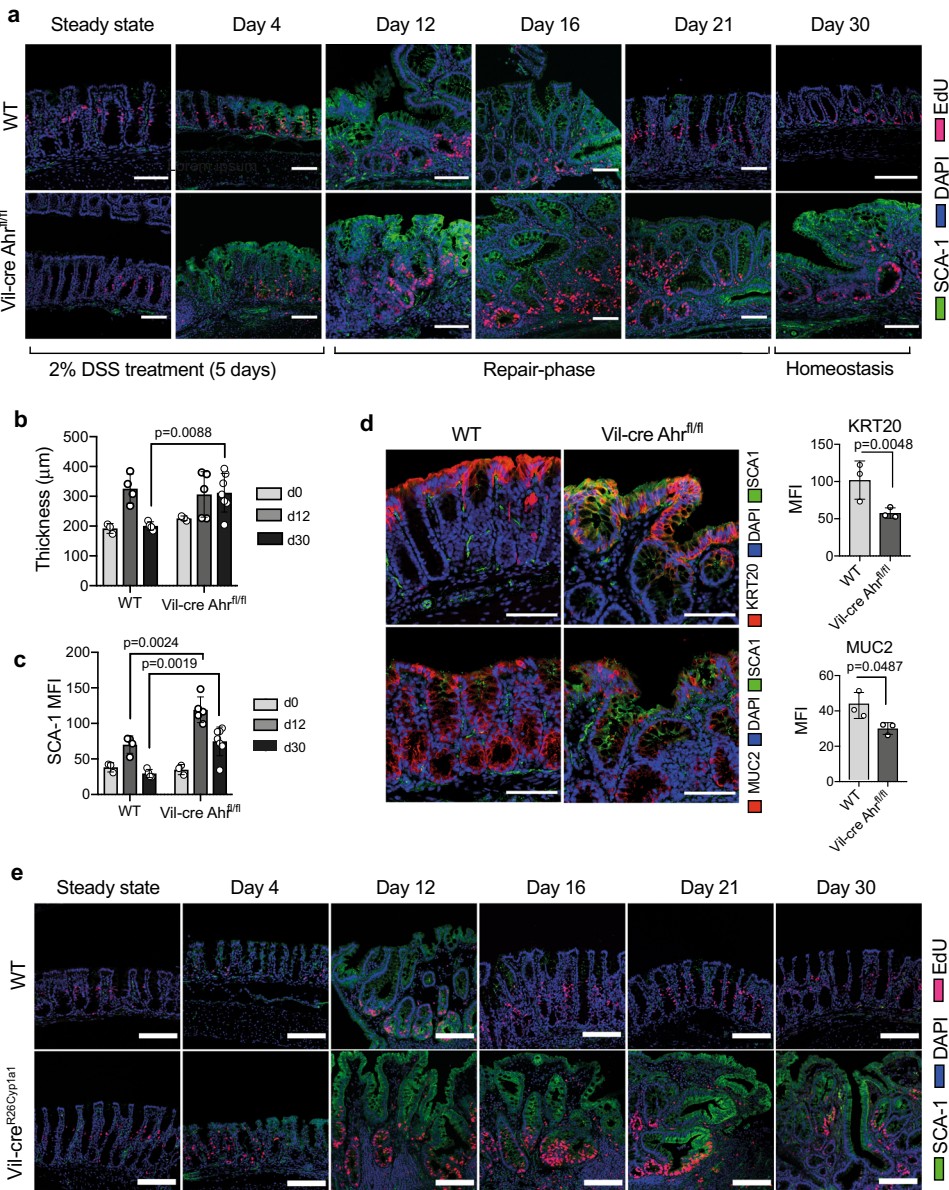

**Fig. 1 AHR drives termination of colonic epithelial regeneration. a** Representative images showing Sca-1 (green) expression and proliferative (EdU+, red) cells in regenerative foci in the colon of DSS-treated Vil-cre AHR$^{fl/fl}$ and WT control mice. Regenerative foci are identified as regions with epithelial Sca-1+ expression and significant crypt dysplasia as well as proliferating cells (EdU; 2 h post-i.p injection). **b** Measurement of mucosal and submucosal thickness and **c** Sca-1 MFI at d0, d12 and d30 post-DSS treatment. (d) Representative images showing expression of differentiation markers Muc2 (goblet cell, red) and Krt20 (pan-differentiation marker, red) in epithelial cells co-stained with Sca1 (green) and DAPI (blue) within regenerative foci in the colon of Vil-cre Ahr$^{fl/fl}$ mice and WT controls at d30 post-DSS challenge. Quantifications for (**b**, **c**) were done measuring mean ± SD distances or mean MFI ± SD per mouse. The data-points represent $n = 3$ (d0), $n = 4$ (d12, d30) WT control mice and $n = 3$ (d0), $n = 5$ (d12), $n = 7$ (d30) Vil-cre AHR$^{fl/fl}$ mice. Statistical significance between genotypes per timepoint was determined by Multiple $t$-test. $P$ value of >0.05 was considered not significant (n.s.). Data-points for (**d**) represents $n = 3$ mice per genotype and quantifications graphed are mean MFI ± SD per mouse; an unpaired $t$-test (two-tailed) was used to assess statistical significance. Representative images show Sca-1 (green) expression and proliferative (EdU+, red) cells in regenerative foci found in the colon of Vil-cre $R26^{LSL-Cyp1a1}$ and WT control mice (**e**) during the course of DSS. DAPI staining (blue) was used to identify nuclei in all images. Data is representative of at least 2–3 independent experiments. Scale bars: 100 μm. Source data for (1**b**–**d**) are provided with this paper in the source data file.

an AHR deficient state[10]. Villin-Cre $R26^{LSL-Cyp1a1}$ mice exposed to DSS exhibited similar functional defects as seen in mice with AHR deficient epithelium with prolonged expression of Sca-1 and failure to terminate regeneration (Fig. 1e). However, in such mice defects in AHR signaling can be mitigated by either increasing the dose of available AHR ligands through the diet[10] or by inhibiting the overactive Cyp1a1 enzyme thereby reinstating the availability of ligands. As shown in Supplementary Fig. 1g, oral application of a potent Cyp1a1/b1 inhibitor (4 L)[23] during the DSS time course

could reverse the detrimental effect of ligand deficiency and reduce Sca-1 expression on day 30 if given during the injury and/or early repair phase. This finding highlights a window within which AHR function is pivotal for the timely resolution of the regenerative response.

**Altered transcriptional program in d4 ENR AHR KO organoids.** Intestinal organoid formation mimics the regenerative response in vivo, and similarly requires transient $Yap1$ activation[21].

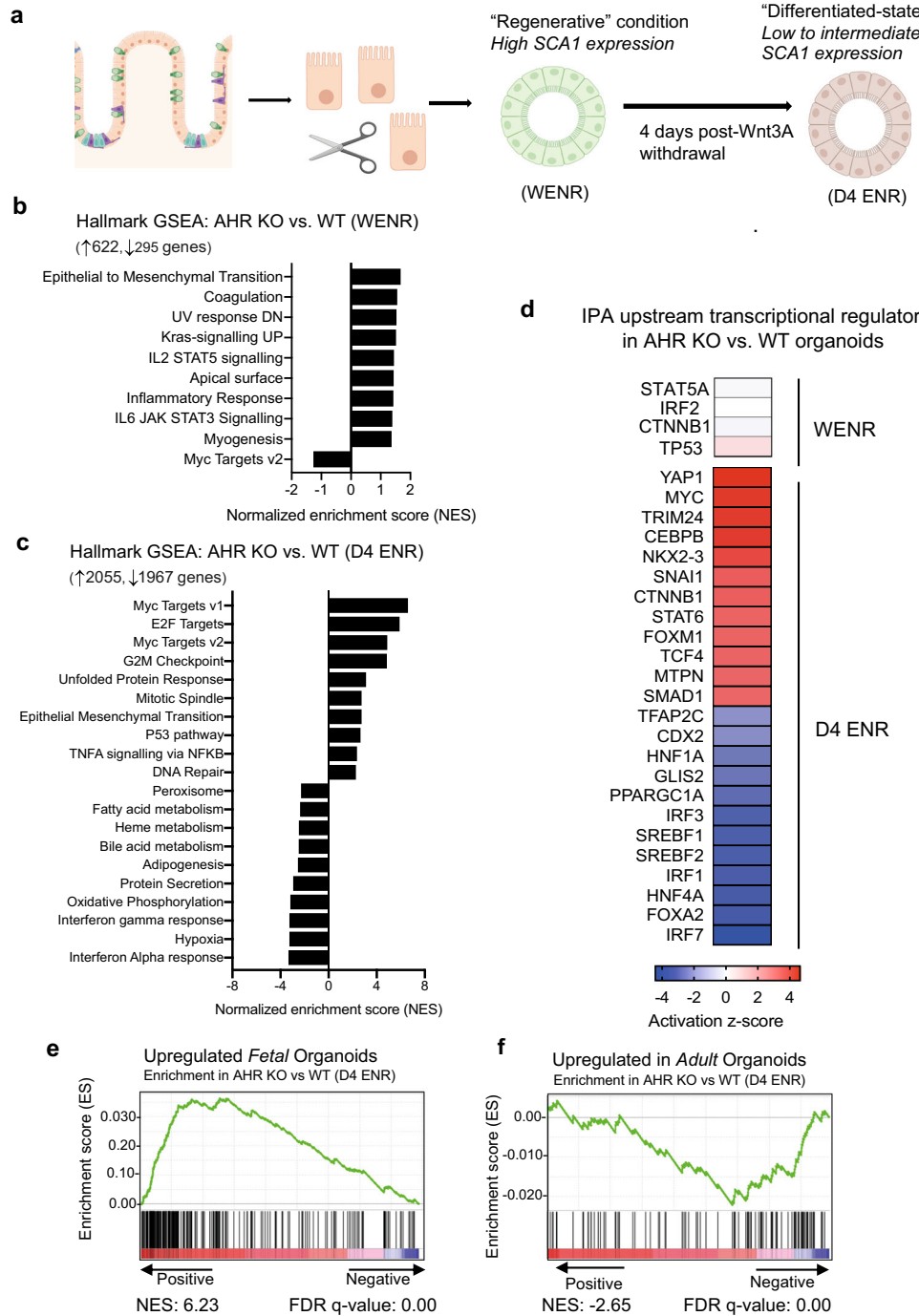

**Fig. 2 Altered transcriptional program in d4 ENR AHR KO organoids. a** Schematic for RNA-seq of organoids in WENR/Regenerative or d4 ENR/ differentiating conditions (created in BioRender). Normalized enrichment scores (NES) from GSEA; MsigDB Hallmark datasets of genes significantly upregulated or downregulated (FDR < 0.05) in AHR KO vs WT colon organoids grown in (**b**) regenerative (WENR) and (**c**) differentiating conditions (d4 ENR). **d** Heatmap for activation z-score of known transcriptional regulators predicted to be upstream of differentially expressed genes in either WENR or d4 ENR AHR KO vs WT organoids (full list in the source data; predicted factors with a p value of <0.0001 are shown). **e** Enrichment plot for transcriptional signature of d4 ENR AHR KO organoids compared to gene set upregulated in (**e**) fetal spheroids (NES: 6.23, FDR q value: 0.00) and (**f**) adult organoids (NES: -2.05, FDR q value: 0.00), respectively. Source data for (2**b**–**d**) are provided with this paper in the source data file.

To gain a deeper understanding of epithelial defects resulting from AHR loss of function, we compared transcriptome profiles of wildtype and AHR-deficient colon organoids grown under regenerative (WENR, Wnt3a supplementation) or differentiating conditions (d4 ENR, four days post-Wnt3a withdrawal (see schematic in Fig. 2a).

In both conditions, AHR KO organoids showed a pronounced alteration in their transcriptional profile in comparison to WT organoids. The number of differentially expressed genes (DEGs) were significantly higher in the d4 ENR conditions (↑2055, ↓ 1967 DEGs) than WENR conditions (↑ 622, ↓ 295 DEGs). Notably, 50–60% of DEGs identified in the WENR condition remained

altered in the d4 ENR condition with many of these genes involved in processes such as cell migration/adhesion and metabolism (Supplementary Fig. 2a). Hallmark gene set enrichment analysis (GSEA) of DEGs identified in AHR KO vs WT organoid grown under WENR conditions showed a modest enrichment for pathways involved in wound repair and epithelial mesenchymal transition (EMT), coagulation or STAT-signalling (Fig. 2b). In contrast d4 ENR AHR KO organoids showed pronounced changes in several pathways (Fig. 2c). Pathways known to be enriched in mature colonic epithelium such as fatty-acid metabolism, peroxisome function, bile acid metabolism were negatively enriched in AHR KO epithelial cells, whereas pathways associated with stemness and injury/regeneration such as those involving *Myc* and *E2F* signalling (proliferation processes), unfolded protein response (ER stress) and epithelial-mesenchymal transition (EMT) were positively enriched (Fig. 2c). These findings highlight AHR involvement in the resolution of the regenerative program and acquisition of mature epithelial identity.

In order to identify regulatory networks influenced by AHR signalling that could account for these changes, we used the IPA Ingenuity software to pinpoint upstream transcriptional regulators predicted to be activated or inhibited in AHR KO organoids in either WENR or d4 ENR conditions. In WENR AHR KO organoids, genes regulated by tumor-suppressor *p53* and *β-catenin* were found to be moderately enriched (positive z-score). In d4 ENR conditions, *Yap1*, along with other proto-oncogenes genes such as *Myc, β-catenin* and *Foxm1*, were predicted to be activated upstream of genes upregulated in AHR KO compared to WT organoids. In contrast, downregulated genes were identified as targets of key TFs involved in intestinal epithelial differentiation such as *Cdx2, Hnf1α* and *Hnf4α* (negative z-score) (Fig. 2d). Yap1 is required for colonic epithelial regeneration both in vivo and in vitro, and its activation results in the transient acquisition of a "fetal-like" transcriptional signature which resolves upon exit from the regenerative program. Through comparison of our d4 ENR data to published datasets generated from fetal-spheroids or mature adult organoids[24] we found that the gene signature in d4 ENR AHR KO organoids bore more similarity to a fetal-spheroid signature (Fig. 2e) whereas the WT organoids exhibited a signature more similar to mature adult organoids (Fig. 2f) further illustrating the failure of AHR KO organoids to terminate the regenerative program.

**Enrichment for Yap/Tead targets in d4 ENR AHR KO organoids.** AHR KO organoids grown in d4 ENR conditions also showed enrichment for a conserved *Yap1* signature, expressing higher levels of canonical *Yap1* transcriptional targets such as *Ctgf and Cyr61* alongside other fetal-like markers associated with colonic epithelial regeneration compared to WT organoids (Fig. 3a, b). This corresponded with higher surface expression of Sca-1 compared to WT organoids (Supplementary Fig. 2b). WNT signaling is required for the initiation of the *Yap*-dependent regenerative program[1,25,26]. In line with the role of AHR in restricting Wnt function, AHR KO organoids also showed increased expression of some canonical WNT targets such as, *Lgr5 and Ascl2* and *Wnt/Yap1* target *Sox9* (Fig. 3c). To determine whether the expression of *Yap1* target genes can be reversed by AHR-signaling, we stimulated WT and AHR KO organoids with the high affinity ligand FICZ under WENR conditions. FICZ-stimulation under regenerative conditions caused the down-regulation of Sca-1 (Fig. 3d) and *Yap1* targets in an AHR-dependent manner (Fig. 3e). Extracellular matrix (ECM) components are key factors that regulate the wound repair process[1]. *Yap1* functions as a mechano-sensor, and upon tissue damage

increased deposition of Collagen I within the wound bed and consequently ECM-stiffness, leads to Yap activation in a *Wnt* and *FAK/Src* dependent manner. To test whether AHR KO organoids are more sensitive to mechano-activation, we grew cells in a matrix enriched for collagen I (60% collagen+ 40% Matrigel). WT and AHR KO organoids grown in d4 ENR conditions responded to mechanical stress by upregulation of *Yap* targets (i.e. *Ctgf*) at the expense of differentiation markers (i.e. *Slc26a3*) (Fig. 3f). Notably, WT organoids grown in 60% collagen expressed these genes at levels similar to AHR KO organoids grown in 0% collagen which may indicate altered mechano-sensing in the absence of AHR. These changes were reversible in WT organoids upon activation of AHR-signaling with AHR agonist FICZ. AHR KO organoids exhibited decreased sensitivity to ROCK inhibition, retaining high expression of YAP targets (e.g. Ankrd1, Ctgf) at a concentration sufficient to abrogate expression in WT organoids grown in 60% Collagen (Fig. 2g). This finding supports the evidence that altered mechano-sensing may be upstream of the changes observed in AHR KO organoids. Collectively, we report that loss of AHR signaling impaired termination of the regenerative state, characterized by sustained expression of Yap targets in differentiating conditions and that *Yap1*-mediated transcriptional regulation can be reversed upon activation of AHR signaling.

**AHR restricts genomic accessibility to Yap/Tead targets.** AHR activation following FICZ-stimulation resulted in the down-regulation of canonical *Yap1* targets, suggesting a direct role for AHR in antagonizing *Yap/Tead*-dependent transcriptional activity. However, our RNA-seq analysis of AHR KO organoids in both WENR and d4 ENR state did not reveal changes in the expression of known regulators of *Yap1* transcriptional activity, and total Yap1 protein levels were largely comparable with WT levels at d4 ENR (Supplementary Fig. 2c, d). Given that global re-wiring of transcriptional networks during intestinal organoid differentiation is preceded and accompanied by mass remodelling of the epigenome[27], we questioned whether AHR might antagonize *Yap/Tead*-mediated transcriptional activity through regulating chromatin accessibility to target genes. To address this, we compared ATAC-seq datasets generated from WT and AHR KO organoids grown in either WENR or d4 ENR conditions and characterized global changes in chromatin accessibility as cells transitioned from a regenerative to a differentiated state. We found that the epigenetic landscape of organoids grown in WENR conditions was comprised of largely open chromatin, whereas that accessibility declined under differentiating, d4 ENR conditions[28] (Supplementary Fig. 3a). Differentially accessible regions between AHR KO and WT organoids grown in d4 ENR conditions (↑1829, ↓415; Fig. 4a) or WENR conditions (↑453, ↓723; Supplementary Fig. 3c) were primarily located in intronic and intergenic regions, suggesting that AHR may influence chromatin accessibility at distal enhancers (Supplementary Fig. 3b). Next, we integrated the d4 ENR ATAC-seq data (AHR KO vs. WT) with RNA-seq data generated from d4 ENR organoids (AHR KO vs. WT) to evaluate whether changes in accessibility correspond with the altered transcriptional profile of AHR KO organoids. Out of differentially accessible genes identified in AHR KO organoids, 22.48% corresponded with transcriptional changes observed in the RNA-seq data (Fig. 4b). Gene ontology analysis of targets identified to overlap between both datasets pinpointed pathways affected by loss of AHR function, corresponding with changes in chromatin accessibility in d4 ENR organoids (Fig. 4b). Genes annotated to differentially accessible peaks overlapping with d4 ENR RNA-seq data were involved in pathways associated with cell motility, migration and tissue

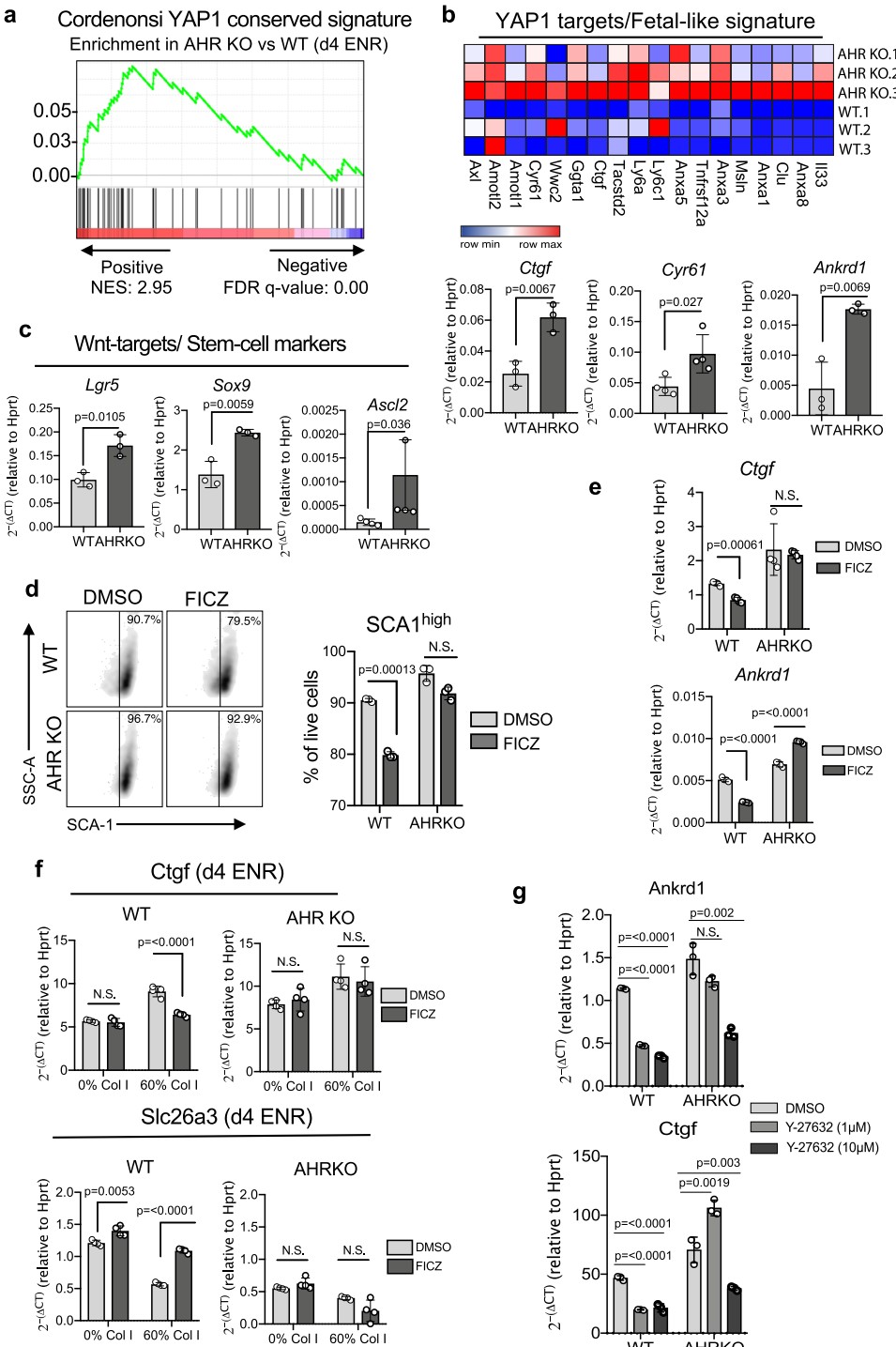

**Fig. 3 Enrichment for Yap/Tead targets in d4 ENR AHR KO organoids. a** Enrichment plot for transcriptional signature of d4 ENR AHR KO organoids compared to the Cordenonsi *Yap1* conserved signature dataset (GSEA C6: Oncogenic datasets - NES: 2.95, FDR *q* value: 0.00) (**b**) Heatmap of select *Yap1* targets and fetal-like genes differentially expressed in d4 ENR AHR KO organoids; expression of YAP1 target *Ctgf*, *Cyr61* and *Ankrd1* validated by qPCR. **c** expression of *Lgr5*, *Ascl2* and *Sox9* by qPCR. **d** Representative flow cytometry plots and percentage of cells expressing high levels of surface Sca-1 (**e**) qPCR expression of *Yap* targets in WT and AHR KO organoids in WENR conditions after 24 h and 4 h of FICZ-stimulation, respectively. **f** qPCR expression of *Ctgf* and colonocyte differentiation marker *Slc26a3* in organoids grown in matrigel containing either 0% or 60% Collagen I. Cells were stimulated with either DMSO (vehicle) or FICZ and expression was assessed 24 h post-treatment. **g** qPCR expression of *Ctgf* and *Ankrd1* in WT and AHR KO organoids grown in 60% Collagen I (d4 ENR) treated with either 1 μM or 10 μM of ROCK-inhibitor (Y-27632). Organoids used for experiments were generated from either *n* = 3 (i.e. *Lgr5*, *Sox9*, *Ankrd1*, *Ctgf*) or *n* = 4 (i.e. *Cyr61*, *Ascl2*) mice per genotype. Data-points for the collagen and Y-27632 experiments are generated from *n* = 4 and *n* = 3 mice per genotype respectively. Statistical significance was determined by (**a**, **b**) unpaired *t*-test (two-tailed) (**c**–**f**) two-way ANOVA with Sidak's multiple comparison test, (**g**) statistical differences between treatment and DMSO control per genotype was done using an ordinary one-way ANOVA with Dunnett's multiple comparison test. *P* value of >0.05 was considered not significant (n.s.). Error bars displayed on graphs represent the mean ± SD of at least two independent experiments. Source data for (3**b**–**g**) are provided with this paper in the source data file.

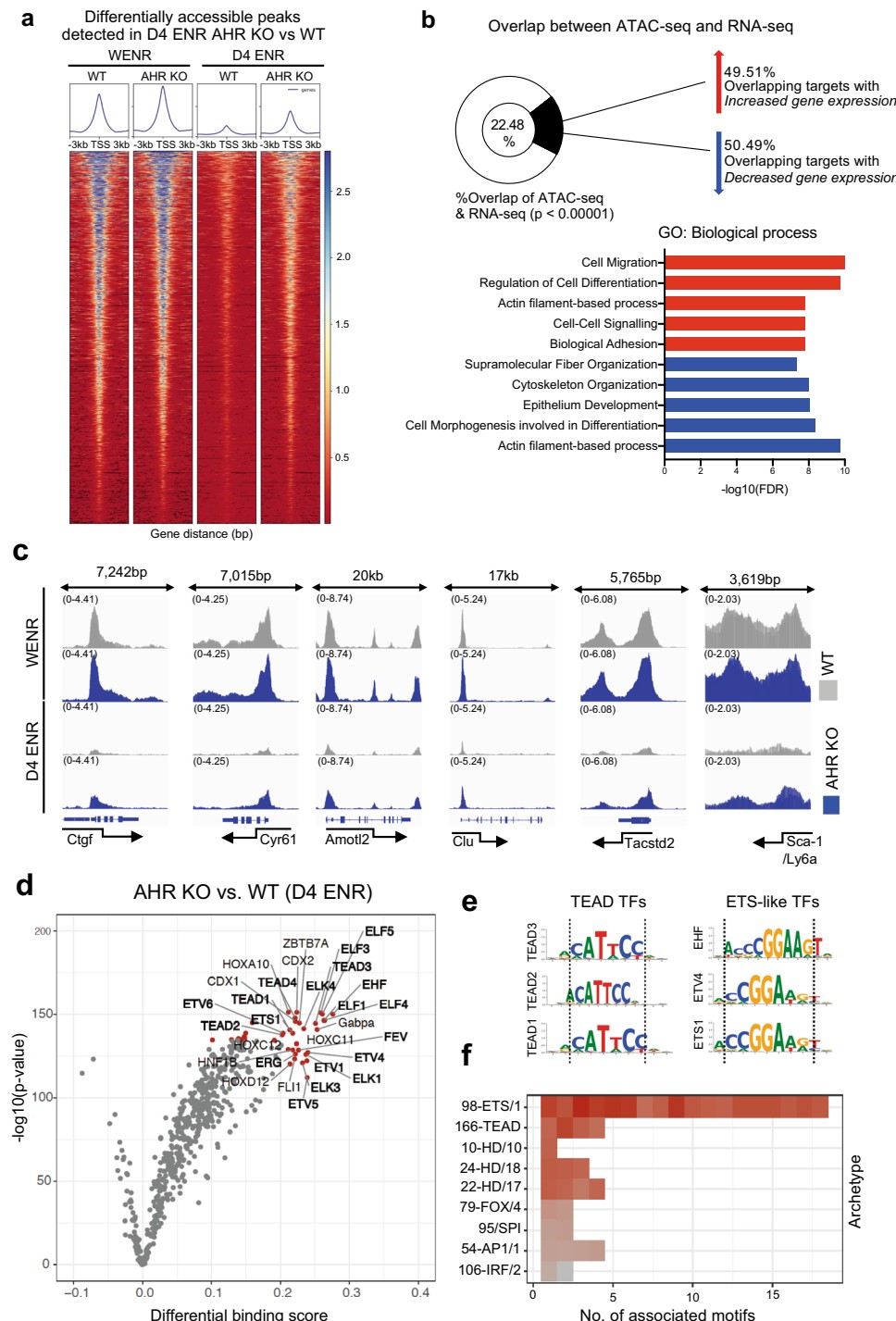

**Fig. 4 AHR restricts genomic accessibility to Yap/Tead targets. a** Deeptools heatmap showing accessibility data of differentially expressed peaks identified in d4 ENR AHR KO vs WT organoids across samples (FDR < 0.05) (**b**) GO gene ontology (biological process) was conducted on genes that overlap between differentially accessible targets identified by ATAC-seq and differentially upregulated or downregulated genes in AHR KO vs WT organoids grown in D4 ENR conditions as determined by RNA-seq (FDR < 0.05). Statistical significance was determined by permutation testing of the overlap between the datasets (p value <0.00001). **c** IgV images of accessibility peaks (open regions) of known *Yap/Tead* targets (*Ctgf, Cyr61, Amotl2*) and fetal-like genes (*Clu, Tacstd2, Ly6a*) in d4 ENR WT and AHR KO organoids (**d**) Pairwise comparison of TF activity between differentiated AHR KO and WT organoids (d4 ENR). The volcano plots show the differential binding activity against the –log10 (p value) of all investigated TF motifs represented by a dot. Only the top 5% (both by fold-change, -log10 p value) of TF motifs identified are highlighted in red. TFs belonging to *Tead* and *Ets-1* like family of factors are bolded. **e** Motif sequences of some identified TFs belonging to *Tead* (*Tead 3, Tead4, Tead1*) and *Ets-1* family are shown, and the dotted line highlights the similarity in the binding sequence of these factors (**f**) Top 5% of identified motifs were aligned to each other and matched to an archetypal consensus motif. The color represents the fold-change (Red to grey – highest to lowest) in differential binding score between d4 AHR KO vs WT organoids. Source data for (4b, d) are provided with this paper in the source data file.

morphogenesis – pathways that are highly regulated by *Yap/Tead* transcription factor (TF) activity[29–31]. Indeed, we found that several *Yap/Tead* targets such as (e.g. *Ctgf, Cyr61, Amotl2*) and fetal-like genes (e.g. *Clu, Tacstd2, Ly6a/Sca-1*) fell into this category of overlapping genes, with significantly increased accessibility in differentiating AHR KO organoids (Fig. 4c). This corresponded with the higher expression of these genes in d4 ENR conditions in comparison to WT controls (Fig. 3b). In contrast, genes involved with cellular differentiation such as *Slc26a3, Muc2, ChgA* did not exhibit differences in chromatin accessibility compared to WT controls (Supplementary Fig. 3d). Thus, AHR may antagonize *Yap1* transcriptional activity by restricting accessibility of its co-activator *Tead* to target genes.

To test this hypothesis, we used TOBIAS, a transcription factor (TF) footprinting tool[32] to predict differential TF occupancy on the genome of AHR KO vs WT organoids as they transitioned from WENR to d4 ENR conditions. In both WT and AHR KO organoids, the differential binding score/transcriptional footprint for *Ap-1* and *Tead*, which drive chromatin opening and *Yap1*-mediated transcription, was notably decreased, whereas pro-differentiation TFs *Hnf4a* and *Rxra*, had an increased transcriptional footprint in d4 ENR organoids (Supplementary Fig. 4a). While both WT and AHR KO organoids seemed to initially share predicted differential binding of TFs during WENR to d4 ENR transition, d4 ENR AHR KO organoids retained a higher differential binding score and footprint for *Tead* factors compared to d4 ENR WT controls (Fig. 4d–f). Differentially accessible sites in AHR KO organoids also exhibited higher scores for *Ets*-like family of TFs upon differentiation which was not evident in WT organoids (Fig. 4d). As multiple TFs can bind similar motifs, we clustered TF motifs based on their archetype group[33], highlighting that many of the identified factors with an increased differential binding score in d4 ENR AHR KO organoids belong to the *Tead* and *Ets*-like family of TFs (Fig. 4e, f).

In the WENR state, *Cdx2* was predicted to have decreased binding in AHR KO organoids, whereas *Ap-1* family transcription factors were predicted to have increased binding in WT organoids (Supplementary Fig. 4b). Collectively, these findings suggest that while both genotypes undergo similar trajectories during the differentiation process, loss of AHR perturbs the overall chromatin landscape, influencing the binding of key TFs that dictate the balance between regenerative and pro-differentiation programs.

**Loss of AHR impairs colonic epithelial differentiation**. AHR has been shown to promote differentiation in a variety of cell types[34,35] and our data indicate that loss of AHR in epithelial cells prolongs the regenerative program at the expense of differentiation (Fig. 2a–e). To assess whether AHR targets specific epithelial cell subtypes or influences the differentiation program in general, we compared our d4 ENR transcriptomic datasets with a published dataset generated from single cell transcriptomic analysis of epithelial cells across the small intestine[36]. The DEGs from d4 ENR AHR KO organoids positively correlated with the transcriptional signature of intestinal stem and progenitor cells, and were negatively enriched for mature epithelial subsets, particularly enterocytes (Fig. 5a). For example, AHR KO organoids expressed considerably lower levels of maturation markers typical for colonocytes (e.g. *Slc26a3, Car4, Alpi*), goblet cells (e.g. *Clca3b, Muc2*) and enteroendocrine cells (e.g. *ChgA*) (Fig. 5b, c). Gene ontology analysis (GO Biological process) also revealed that many downregulated genes in d4 AHR KO organoids are involved in key functions of mature intestinal cells such as absorption and digestion (Fig. 5d, e). Intestinal organoids undergo significant metabolic rewiring as they transition from WENR state to differentiating ENR conditions, characterized by increasing metabolic reliance on oxidative phosphorylation at the expense of glycolysis[37]. To get an additional insight into how AHR regulates the differentiation process, metabolic features of stem cell and differentiating colonic organoids were assessed. WT organoids underwent a metabolic shift indicated by an increasing OXPHOS/Glycolysis ratio (Fig. 5f) and a concomitant decrease in the glycolytic index following the initiation of the differentiation process. In contrast, AHR KO organoids showed sustained preference for glycolysis for its energy demands (Fig. 5f) and impaired ability to undergo a switch to increased OXPHOS reliance under differentiating conditions. This finding is complementary to the decreased enrichment of d4 ENR AHR KO organoids for genes associated with oxidative phosphorylation (Fig. 2b).

**AHR regulates key factors involved in regenerative response**. AHR functions as a ligand-dependent transcription factor in mammalian cells, and we hypothesized that AHR may regulate the exit from the regenerative state and drive differentiation through transcriptional regulation of key factors involved in this process. To address this, we conducted CHIP-seq analysis of FICZ treated WT and AHR KO organoids to identify ligand-dependent AHR-specific targets under WENR conditions. Through this analysis, we identified 121695 peaks bound by AHR annotated to 18416 unique gene targets. To focus on targets most likely to have a biologically relevant outcome in our system, we restricted our analysis to binding sites in epigenetically accessible regions of the genome that would identify AHR targets primed for activation or repression in the WENR state. We therefore overlapped our ATAC seq dataset from WT organoids grown under WENR conditions with our CHIP-seq data. Our analysis identified 3662 targets annotated to 2785 unique genes, with AHR binding occurring in regulatory elements within intragenic (43.8%) and intergenic (42.1%) regions similar to what has been described in other AHR-CHIP datasets (Supplementary Fig. 5a), and 79% of targets near protein coding genes (Supplementary Fig. 5b)[38,39]. HOMER motif analysis of these AHR-bound regions revealed the *Ahr:Arnt* motif as the most enriched, followed by the *Ap-1* motif (Supplementary Fig. 5c). Next, we integrated this dataset with transcriptomic data generated from RNA-sequencing of WT and AHR KO organoids stimulated with FICZ (4-hours post-stimulation; WENR) to further refine our analysis to AHR-dependent targets with a transcriptional outcome. Using this approach, we found that 21.42% or 19.63% respectively of genes upor downregulated in the RNAseq dataset corresponded with 13.2% of AHR ChIP targets in open chromatin (Fig. 6a). Collectively we labelled AHR targets identified from the integration of these datasets as "active" AHR targets. Gene ontology analysis revealed enrichment for several pathways involved in epithelial differentiation and tissue/structural morphogenesis in genes activated upon AHR activation, whereas genes involved in the regulation of cell migration and biological adhesion processes were downregulated (Fig. 6b). Notably, we identified *Cdx2*, a key transcription factor involved in intestinal epithelial differentiation and specification, as an active AHR target, alongside other transcription factors known to amplify and/or crosstalk with *Cdx2* in the intestine such as *Cdx1, Hnf1a*[40–42] and *Rxra*[20] (Fig. 5c). Conversely, *Sox9* (a *Wnt* and *Yap1*-target gene) and some canonical targets of *Yap/Tead* such as *Cyr61* and *IL33* which have known roles in stem cell maintenance, cell migration and tumor invasiveness - were repressed by AHR (Fig. 6c). AHR bound to an annotated enhancer element downstream of the *Cdx2* gene (OREG1865271; PAZAR) involved in positive regulation of *Cdx2* (Fig. 6d, f). In contrast, AHR bound to the *Sox9* promoter and

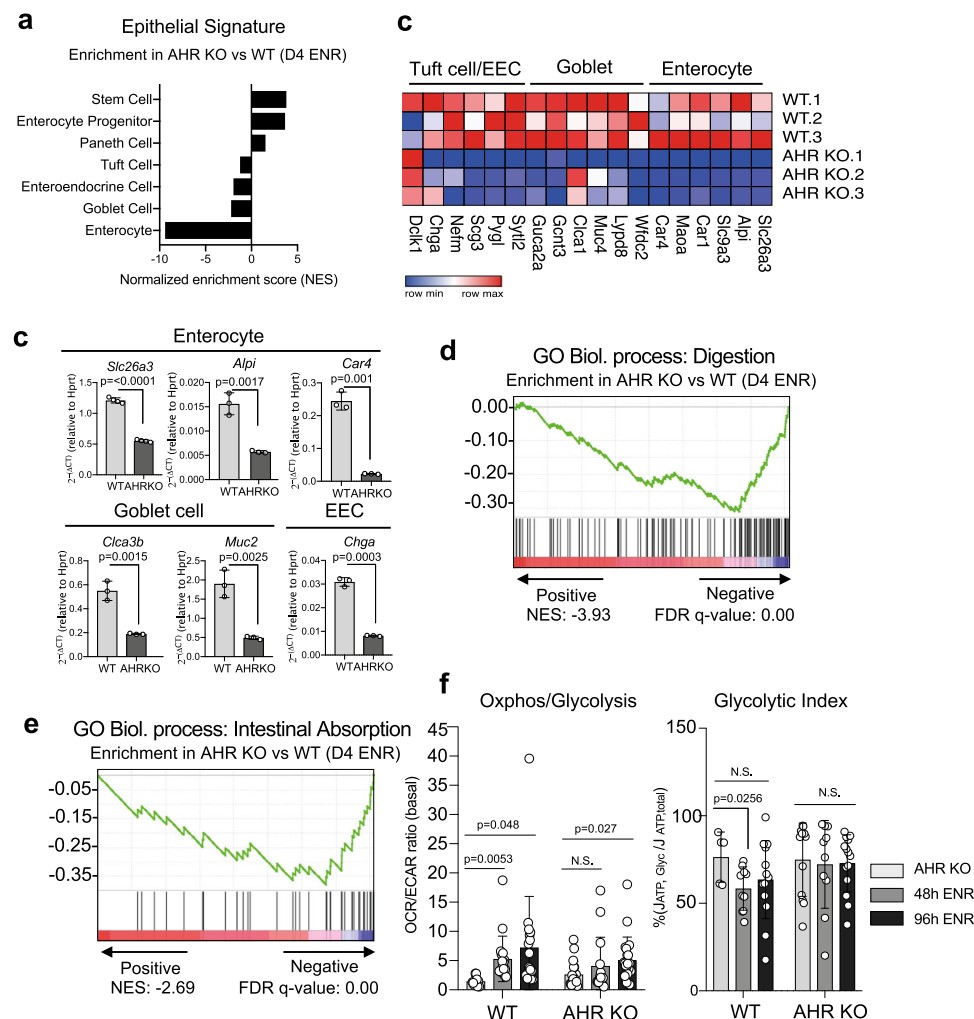

**Fig. 5 Loss of AHR impairs colonic epithelial differentiation. a** Enrichment plot for transcriptional signature of d4 ENR AHR KO organoids compared to transcriptional signatures from intestinal epithelial cell subtypes. **b** Heatmap showing expression of select genes for mature epithelial subsets in either WT or AHR KO cells grown in d4 ENR conditions (**c**) qPCR expression data of mature epithelial markers for colonocytes (*Slc26a3*, *Alpi*, *Car4*), goblet cells (*Clca3b*, *Muc2*) or enteroendocrine cells (*Chga*) in d4 ENR WT or AHR KO cells. Organoids used for experiments in (**c**) were generated from either $n = 3$ (i.e. *Muc2*, *ChgA*, *Clca3b*, *Car4*) or $n = 4$ (i.e. *Slc26a3*) mice per genotype and statistical significance was determined by unpaired *t*-tests (two-tailed). Error bars displayed on graphs represent the mean ± SD of at least three independent experiments. **d** GO gene ontology analysis for transcriptional signature associated with mature epithelial function such as digestion (NES: −3.93, FDR *q* value: 0.00) and (**e**) absorption (NES: −2.21, FDR q-value: 0.00). **f** Metabolic analysis by Seahorse of AHR KO organoids compared with WT controls for basal OxPHOS vs glycolysis (left panel) and glycolytic index (right panel) and assessed at either 0 h (WENR conditions), 48 h (d2 ENR) and 96 h (d4 ENR) post-Wnt removal. Data shows mean ± SD expression of technical replicates from organoids generated from $n = 3$ mice and data pooled from three independent experiments. Multiple *t*-tests was performed between each timepoint vs WENR condition per genotype to determine statistical significance. Source data for (5**c**, **f**) are provided with this paper in the source data file.

repressed its expression (Fig. 6e, f). Notably, the promoter region of Sox9 contains YAP/TEAD binding sites[43] in close proximity to several AHR binding sites, suggesting that these factors may counter-regulate Sox9 expression directly (Supplementary Fig. 5d). Based on this observation, we investigated whether other differentially accessible regions might be co-regulated by both AHR and YAP/TEAD by subsetting the ATAC peaks with overlapping AHR chip signatures. We found 294/2241, and in these 294 ATAC regions, 73/294 contained both *Ahr* and *Tead* motifs that co-localised in the same open ATAC region. 998 pairs of *Ahr* and *Tead* motifs were within the same open ATAC region with the majority of the distances < than 1Kb away from each other and primarily characterised as intergenic, intronic or promoter regions. We saw weighting towards intergenic and intronic regulatory elements suggesting a possible bias towards enhancer activity over direct TSS activity (Supplementary Fig. 6a, b). GSEA (GO Biological Process) for the nearest gene IDs of these 73 peaks

(Supplementary Fig. 6c) revealed enrichment for processes involved in the regulation of cell motility and adhesion, in line with the proposed function of *Ahr* and *Yap/Tead* signalling in the co-regulation of these processes. Some examples of genes in this category are Sox9, Itga2 and Nrp2 (Supplementary Fig. 6d). In conclusion, there are numerous examples where AHR and Tead are co-localising in the same differentially accessible ATAC region in functional areas such as enhancers and promoters. Thus, in addition to restricting chromatin accessibility to *Yap/Tead* targets, AHR orchestrates the termination of the regenerative state through direct transcriptional regulation of genes involved in intestinal epithelial differentiation, and repression of genes involved in stem cell and wound-repair associated programs.

**AHR drives reacquisition of intestinal identity post-injury.** *Cdx2* is required for differentiation towards mature epithelial cell

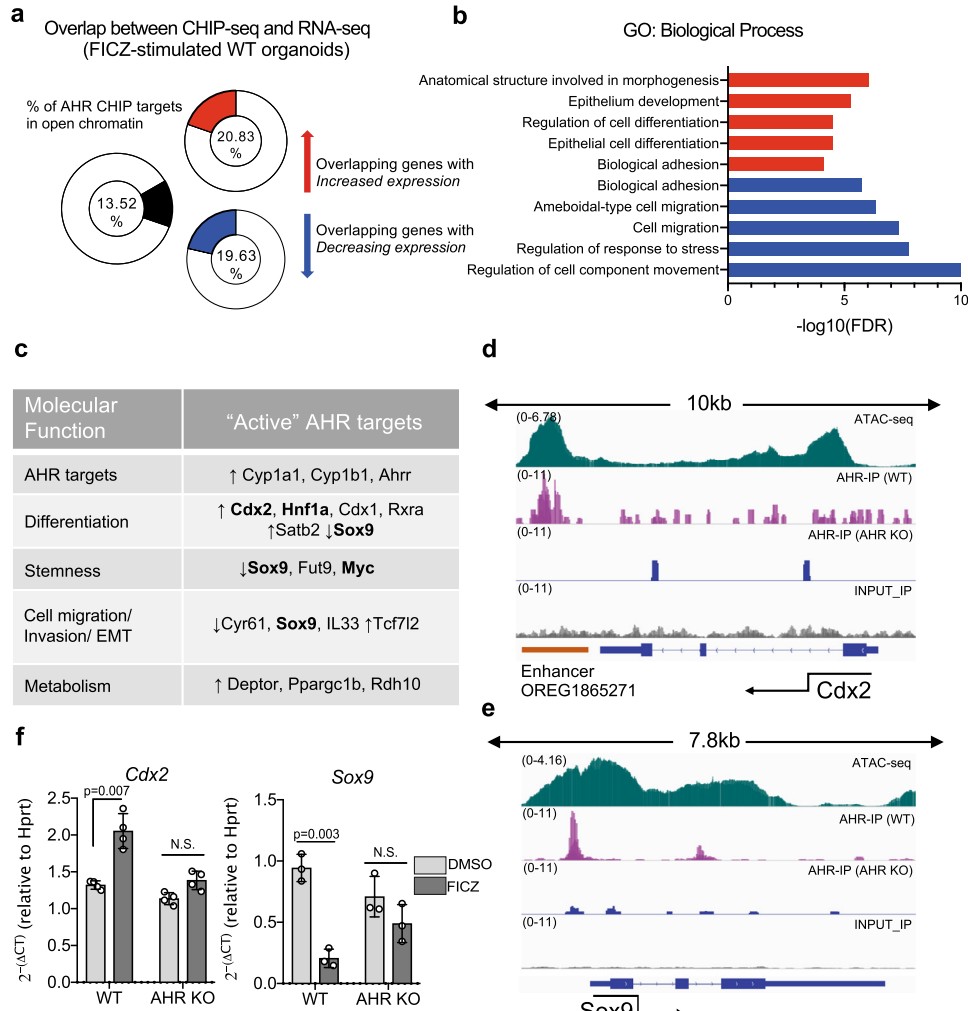

**Fig. 6 AHR regulates key factors involved in regenerative response. a** Overlap between identified AHR ChIP targets in open chromatin regions with genes up or downregulated in response to FICZ from RNA-sequenced WT organoids (4h-post FICZ treatment). **b** Functional annotation of activated (red) or repressed (blue) AHR targets using GO ontology analysis (GO biological process) (**c**) Selection of active AHR targets with their associated functions. Targets that were also identified by the IPA upstream regulator analysis in d4 ENR AHR KO organoids are in bold (see Fig. 2 C). **d** IgV graphs showing sites of AHR binding in an enhancer element of *Cdx2* and and (**e**) AHR binding to the *Sox9* promoter. **f** qPCR expression data of *Cdx2* and *Sox9* in WT and AHR KO organoids treated with FICZ (4 h). Statistical significance was determined by two-way ANOVA with Sidak's multiple comparison test. *P* value of >0.05 was considered not significant (n.s.). $n = 4$ and $n = 3$ mice per genotype were used for *Cdx2* and *Sox9* qPCRs respectively. Error bars displayed on graphs represent the mean ± SD of at least two independent experiments. Source data for (6**a**–**c**, **f**) are provided with this paper in the source data file.

fates––especially enterocytes and *Muc2* expressing goblet cells[41,44–46]. Epithelial-specific deletion of *Cdx2* results in the anteriorization of the intestine, causing the expression of gastric-like genes[47]. Given that Cdx2 is positively regulated by AHR (Fig. 6d, Fig. 2c), along with TFs such as *Hnf1a* and Rxra which synergize with *Cdx2* to drive intestinal epithelial differentiation, we assessed whether loss of AHR corresponds with changes due to *Cdx2* deficiency by comparing the transcriptional signature of d4 ENR AHR KO and Cdx2 KO organoids. We observed a strong positive correlation between both datasets, with the majority of genes up or downregulated in Cdx2 KO organoids corresponding with changes observed in the transcriptional signature of AHR KO organoids (Fig. 7a). In addition, the transcriptional signature of AHR KO organoids was negatively enriched for the expression of genes characteristic for the small intestine and traverse/prox-imal colon where functions such as absorption and generation of mucins for barrier protection occur[48]. Instead, AHR KO orga-noids exhibited positive enrichment for genes expressed in anterior regions of the GI tract such as oesophagus and stomach and for the sigmoid colon which represent regions with decreased

*Cdx2* activity (Fig. 7b)[49]. Furthermore, we observed increased expression of gastric genes and decreased expression of intestinal genes (Fig. 7c) in d4 ENR AHR KO organoids compared with WT controls. Expression of *Cdx2* and *Sox9* are inversely corre-lated during organ specification and colorectal cancer and *Sox9* is a known repressor of *Cdx2* in the intestine[44,50,51]. In line with this, regenerative foci retained in Vil-cre AHR^fl/fl at d30 post-DSS exhibited decreased Cdx2 staining and increased Sox9 staining (Fig. 7d, e). Collectively, these data demonstrate significant par-allels between AHR KO and Cdx2 KO phenotypes and provide evidence for AHR as a key factor driving the re-establishment of intestinal identity.

**Discussion**
In this study we provided evidence for an important role of the environmental sensor AHR in coordinating processes involved in tissue repair and differentiation. Using an integrated multi-omics approach in colon organoids, we reveal a requirement for AHR in the appropriate termination of the regenerative response via antagonizing

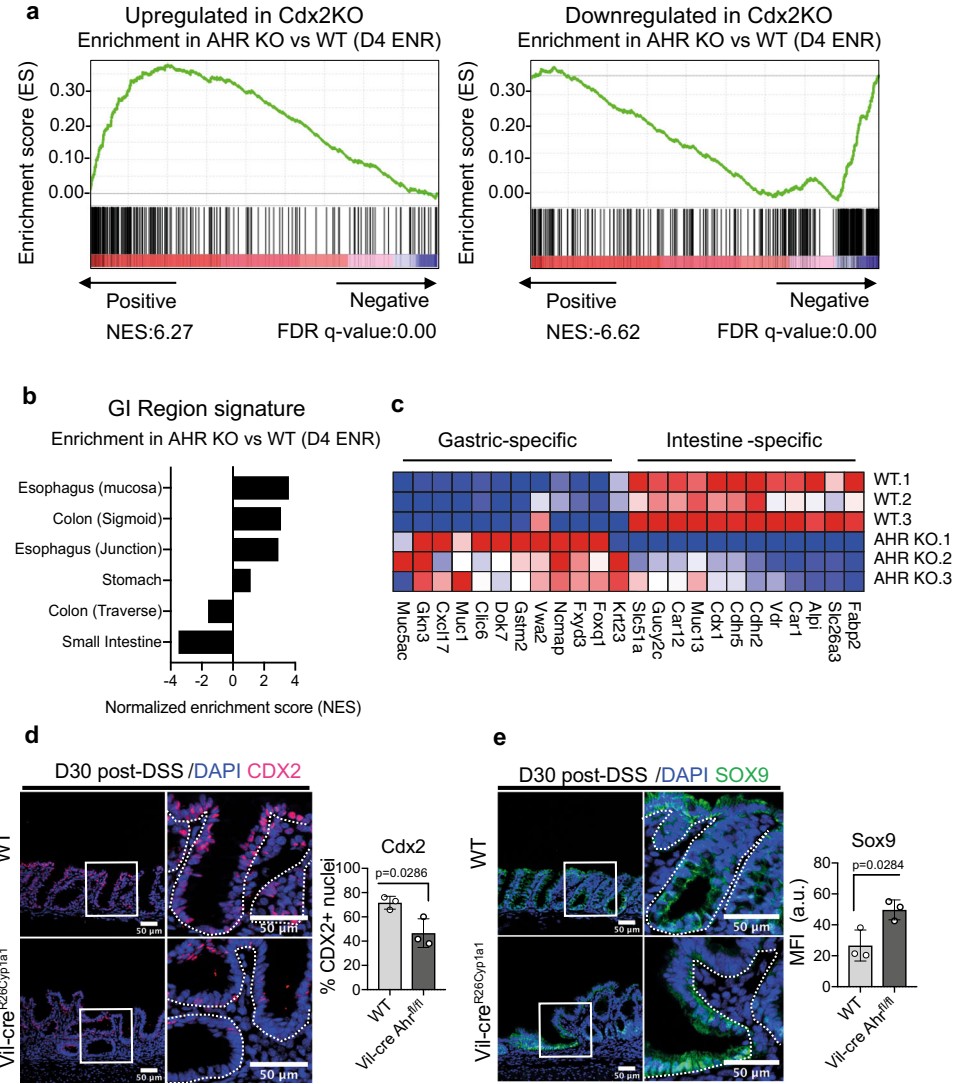

**Fig. 7 AHR drives reacquisition of intestinal identity post-injury. a** Gene set enrichment plots of AHR KO organoids relative to genes either upregulated (NES: 6.3, FDR q-value: 0.00) or downregulated (NES: -6.62, FDR q value: 0.00) in Cdx2 KO intestinal organoid dataset (**b**) Gene set enrichment of d4 ENR AHR KO vs WT organoids relative to conserved GI-region specific signatures compiled from GiTEX database (**c**) Heatmap for expression of canonical genes expressed in either intestine or gastric tissues, in d4 ENR WT and AHR KO organoids. Representative images and quantifications of Cdx2 (**d**) and Sox9 (**e**) staining in colonic crypts of WT and Vil-cre AHR^fl/fl mice d30 post DSS treatment. Bar graphs show mean MFI ± SD per mouse for Sox9 and mean % of nuclei positive for Cdx2 per mouse (n = 3 mice). Statistical significance was determined using an unpaired t-test (two-tailed). Error bars displayed on graphs represent the mean ± SD. Scale bar: 50 μm. Source data for (7**d**, **e**) are provided with this paper in the source data file.

*Yap/Tead*-mediated transcriptional regulation of target genes in favour of a pro-differentiation program through its regulation of master intestinal factor *Cdx2*. We found that AHR functions as a transcriptional activator of pro-differentiation factors (e.g. *Cdx2, Hnf1a, Rxra*) and repressor of pro-stemness factors (e.g. *Sox9, Myc*).

An association between AHR and *Cdx2* was previously described by us in the context of embryogenesis, where activation of AHR in embryonic stem cells resulted in the early induction of *Cdx2* under differentiating conditions[34], but whether AHR could regulate *Cdx2* under physiological conditions remained unknown. In this study, we established *Cdx2* as a direct AHR transcriptional target in colonic epithelial cells. The strong correlation between the transcriptional signature of AHR KO and Cdx2 KO organoids provided further support for the requirement of AHR in the optimal induction of *Cdx2* and consequently the reestablishment of mature intestinal epithelial identity post-regeneration. *Sox9* was also a factor of interest as the antagonism between AHR and *Yap/Wnt*

signaling converges on its regulation, and it is involved in the maintenance of stem-cell and EMT promoting transcriptional networks. Furthermore, a bi-directional relationship between Sox9 and Cdx2 has been demonstrated in the context of malignancy, with *Sox9* highly expressed in metaplastic regions that have lost Cdx2 expression, illustrating that Cdx2 restricts Sox9 activity[44]. Conversely, *Sox9* can also repress *Cdx2* expression[51]. Therefore, it is possible that sustained Sox9 activity may further contribute to the reduction of *Cdx2*-driven intestinal epithelial specification/differentiation we see in AHR deficient epithelium.

Previous studies have provided evidence for AHR-signaling in the repression of the canonical Wnt-signaling (via β-catenin)[19,22,52] and Foxm1 signaling[53], but whether *AHR* antagonizes other pro-regenerative/oncogenic pathways is still a subject of active research. AHR-mutants have increased susceptibility to colorectal cancer development[19,22] and our work provides further mechanistic insight into the cancer protective functions of AHR in colonic

epithelial cells by demonstrating a key role for AHR in restricting *Yap1*-mediated transcriptional activity. Dysregulated *Yap*-signaling is associated with tumorigenesis in various tissues[25,31,54–56].

WENR organoids, which resemble injury-induced "fetal-like" stem cells exhibit increased chromatin accessibility, with open regions that are highly enriched for *Ap-1* and *Tead* motifs. While the transient reprogramming of cells to this reverted state has the important function of increasing the stem-cell pool to drive rapid regeneration, its perpetuation may lead to an increased likelihood for oncogenic transformation if it occurs in the presence of mutagens. This is particularly relevant for barrier sites such as the intestine which is chronically exposed to external factors and is injury-prone[2,3,57]. It is noteworthy that while the loss of AHR function under homeostatic conditions did not have overt implications on epithelial turnover or differentiation under steady-state conditions, we cannot rule out pre-existing vulnerabilities in the chromatin that could make AHR-deficient cells more susceptible to injury-induced repair processes. This feature of AHR function is unlikely to be shaped by differential availability of ligands in steady state vs injury as tryptophan is an essential amino acid and AHR ligands are constantly produced from dietary intake as well as the microbiota. The context-dependent function of AHR is reminiscent of *Yap1* activity, which is likewise found to be largely dispensable under homeostatic conditions but is required for the initiation and propagation of the regenerative response[1,25,58,59]. We propose that AHR may increase the threshold for oncogenic transformation by limiting *Yap1*-mediated transcriptional activity.

The more open chromatin configuration of AHR KO epithelial cells during differentiation could reflect the increased representation of *Ap-1* which is involved in regulating chromatin dynamics[60]. Indeed, AHR activation can restrict *Ap-1* activity in a cell -type dependent manner[61–63]. Furthermore, we found that *Ap*-1 is the second most common motif in AHR-bound regions identified in the ChIP analysis, suggesting an antagonistic interplay between *Ap-1* and AHR in regenerating cells. There is increased footprinting of *Ap*-1 complex members in AHR KO organoids – most notably in the WENR state which may reflect the higher sensitivity of AHR KO organoids to *Ap-1*-mediated opening of chromatin. *Ap-1* associates with *Yap* to promote the expression of motility genes, and *Sox9* in the expression of ECM-associated genes[31,56,60,64–66]. *Ap-1* binding sites are also commonly associated with the binding of *Ets*-like TFs which may explain the differential increased binding score of this family of factors in differentiating AHR KO organoids. Thus, loss of AHR resulting in dysregulated *Ap-1* activity might contribute to the persistence of *Yap/Tead* activity and facilitate the activity of *Ets*-like transcription factors.

The gene ontology analysis of our integrated genomics also suggests a role for AHR in the regulation of pathways involved in motility/ migration and biological adhesion in colonic epithelial cells both at a transcriptional and epigenetic level. The persistence of *Yap/Tead* and *Sox9* in d4 ENR AHR KO organoids may largely account for these transcription signatures, as they actively promote pro-metastatic programs through regulation of cellular adhesion and motility pathways which are enriched for genes that encode various ECM-components and factors that influence the actin cytoskeleton. Our findings are in line with previous studies demonstrating that reduced AHR leads to increased cell invasiveness in certain cancers[67,68]. Furthermore, AHR activation has also been shown to protect against tissue fibrosis[69,70]. However, opposing findings have been reported by other studies which have shown that AHR can promote tumor invasiveness, but these findings are reported in the context of AHR activation by TCDD which could have adverse outcomes[71]. The transient nature of AHR activation under physiological conditions by dietary and microbiota-derived ligands highlights the importance of controlled AHR activity. Indeed, chronic activation can also lead to ligand-depletion through excessive activation of negative feedback pathway which leads to a quasi-AHR deficient state[10,72].

Collectively, our work provides mechanistic insight into AHR-mediated functions in the context of colonic tissue regeneration, which when dysregulated, could lead to CRC initiation and progression. AHR is an environmental sensor that integrates microbial and dietary-cues to influence physiological/cellular processes within the intestinal microenvironment and beyond. The growing link between environmental factors such as dietary intake, pollutants and microbial dysbiosis in colorectal cancer etiology, places AHR at a pivotal position in influencing the delicate balance between controlled regeneration and malignant transformation.

## Methods

**Mice**. Villin^CreAhR^fl/fl (AhR tm1c generated from AhR tm1a ES cells from KOMP), Villin-Cre R26^LSL-Cyp1a1^(10) and their wildtype littermate controls (all on a C57BL/6 background) were bred and maintained in individually ventilated cages at the Francis Crick Institute, under specified pathogen free (SPF) conditions according to the protocols approved by the UK Home Office and the Ethics committee of the Francis Crick Institute. Littermates of the same sex were randomly assigned to experimental groups.

**DSS colitis**. For induction of DSS colitis, mice were treated with 2% DSS solution (MP Bio, Cat: 0216011090) in drinking water for 5 days followed by normal water. To track cell proliferation within the crypt compartment, mice were injected (i.p.) with 200µg of EdU (Thermofisher; Cat: A10044) for 2 h prior to harvesting of tissues.

**Histology and immunostaining**. Colonic tissues were fixed in 4% Paraformaldehyde (PFA; Thermofisher, Cat: 28908) for 3 h (for OCT embedded samples) or 24 h (for paraffin embedded samples) at 4 °C. For frozen sections, samples were incubated in a 10% glycerol 25% sucrose solution overnight after fixation, prior to embedding in OCT. OCT sections were used for the rat anti-Sca-1 (1:200; Biolegend, Cat: 122501), rabbit anti-Cdx2 (1:300; Abcam, Cat: ab76541), rabbit anti-Muc2 (1:200; Abcam, Cat: ab272692), rabbit anti-Krt20 (1:200; Cell Signalling, Cat: 13063) and rabbit anti-Sox9 (1:200; Millipore, Cat: AB5535) for IF staining. For IHC/IF staining on paraffin embedded samples, sections were deparaffinised and rehydrated using standard methods and were stained with either rabbit anti-Dcamkl1 (1:200; Abcam, Cat: ab31704) or rabbit anti-ChgA (1:500; Abcam, Cat: ab15160). Citrate buffer pH6 was used for antigen retrieval and tissues were stained with primary antibodies overnight after incubation with blocking solution for 2 h, and EdU staining with the Click-IT EdU kit (Thermofisher; Cat: C10337) following manufacturer's instructions. Secondary antibodies (all from Thermofisher used at 1:500) - Donkey anti-Rabbit IgG (H + L) Highly Cross-Adsorbed Secondary Antibody (AF568 Cat: A10037 or AF647 Cat: A31571), Donkey anti-Rat IgG (H + L) Highly Cross-Adsorbed Secondary Antibody (AF488 Cat: A48269) and DAPI (1:2000; Thermofisher, Cat: D1306) were incubated for 2 h at room temperature. Fluorescent images were acquired using a laser scanning confocal microscope (LSM710 Zeiss Upright). For quantifications of Sca1, Muc2, Cdx2 and Krt20 MFI from IF sections, several images were acquired from the mid-distal region of the colon. In general, images were acquired from regenerative regions (indicated by epithelial Sca1 expression). However, in samples without Sca1+ regenerative foci (i.e. steady-state conditions and WT samples post-regeneration), images of crypts were also acquired from the mid-distal colon. Images were obtained from biological replicates across 2–4 independent experiments and the average MFI (after subtraction of background MFI) per image and sample was plotted. Ilastik (pixel classification tool; version 1.3.3.post)[73] was used for cell segmentation and classification of DAPI+ and Cdx2+ from colon images. For quantification of % Cdx2+ cells per nuclei––the number of identified Cdx2+ cells were normalized to the number of DAPI+ cells within the colon crypt. All images were quantified using ImageJ version 2.1.0/1.53c.

**Primary culture of mouse colon organoids in Matrigel or Collagen type I**. Colonic crypts were isolated from the whole colon of WT and *Vil*^CreAhr^fl/fl mice (n = 3–4, biological replicates) in a PBS solution containing 5 mM EDTA. The tissue was incubated for 30–45 mins in a 37 °C shaking incubator (200RPM), followed by washing and manual shaking in cold D-PBS to isolate crypts. To generate organoids, isolated crypts were embedded in Matrigel (R&D systems Cat: 3533-010-02; 70% Matrigel mixed with 30% WENR media) and resulting organoids were maintained in either WENR medium or switched to ENR only medium for four days to induce differentiation. As previously described (Metidji A et al, 2018), the basic culture medium (ENR) contained advanced DMEM/F12 supplemented with penicillin/streptomycin, 10 mM HEPES, 2 mM Glutamax, B27 (all from Life Technologies) and 1mM N-acetylcysteine (Sigma; Cat: A7250) supplemented with

murine recombinant EGF (0.5 µg/ml; Gibco cat: PMG8043). R-spondin1-CM (20% v/v) (20% final volume) and Noggin-CM (20% v/v) were obtained from the Cell Services platform at the Francis Crick Institute. The 'WENR' medium was generated by using ENR as a base medium with 50% WNT3a-conditioned medium (obtained from Cell Services at the Francis Crick Institute) supplemented with SB202190 (10 µM; Sigma, Cat: S7067-5MG), ALK5 inhibitor A83-01 (500 nM; Tocris Bioscience, Cat: 2939) and nicotinamide (10 mM, Sigma: Cat: N3376). For the experiments involving use of organoids grown in Collagen Type I (Rat Collagen I 5 mg/ml, RND systems Cat: 3447-020-01), passaged organoids were embedded a Collagen matrix diluted in Matrigel to a final concentration of 3 mg/ml (60% Collagen I; 40% Matrigel). For experiments using ROCK-inhibitor Y-27632 (Stemcell technologies, Cat: 72304) organoids were stimulated with either at 1 µM or 10 µM at D3 ENR and harvested for analysis at D4 ENR.

**Flow cytometry and Western blotting**. For flow cytometric analysis or cell-sorting for ATAC-seq, single-cell suspensions were generated by resuspending organoids in TrypLE (Gibco, Cat: 12604013) supplemented with DNAse-I followed by gentle dissociation by pipetting, and incubation in a 37 °C water bath for a total of 15 mins. Cells were stained with fixable Near IR live/dead viability dye (1:1000; Thermofisher Cat: L10119) for 20–30 mins on ice. For surface staining, cell suspensions were incubated with BV421 anti-Sca-1 (1:200; BDBiosciences Cat: 744322) for 30 mins on ice. Cells were acquired with a BD Fortessa Cytometer and analysis was performed with FlowJo (Tree Star) software. Colon organoids single-cell sorted for ATAC-seq, were sorted on a FACS Fusion Aria II. For quantification of total Yap1 and ß-actin levels, organoids grown under WENR, 48 h ENR and 96 h ENR were harvested with TrypLE (Gibco, Cat: 12604013) and washed twice with PBS prior to lysis with RIPA buffer supplemented with protein inhibitor cocktail (1:100; Cell Signalling Technologies, Cat: #5871). Total protein levels were quantified with a Pierce™ BCA protein assay kit (Thermofisher, Cat: 23225) and 20 µg of protein were loaded and separated in 4–15% Mini-PROTEAN® TGX™ Precast Protein Gels (Bio-rad, Cat: #4561084) and transferred into nitrocellulose membranes. Membranes with blocked with 5% BSA in TBST for 1 h incubated with both rabbit anti-Yap XP (1:1000; Cell Signalling Technologies, Cat: #14074) and ß-actin (1:2500; ThermoFisher, Cat: # MA1-91399) antibodies overnight. For detection, washed membranes were incubated with LI-COR Biosciences antibodies (IRDye800CW Donkey anti-rabbit IgG (H + L) Cat: 925-32213; IRDye680LT Donkey anti-mouse IgG (H + L) Cat: 925-68022; both used at 1:10000) in blocking buffer for 1 h. Chameleon® Duo Pre-stained Protein Ladder (Li-Cor Biosciences, Cat: 928-60000) was used as a molecular weight marker. Signal from Blots were measured using a LI-COR Odyssey CLx (LI-COR, Lincoln, USA) system.

**Seahorse XF metabolic analysis**. Seahorse Bioscience XFe96 Analyzer was used to measure extracellular acidification rates (ECAR) in mpH (milli pH) per min and oxygen consumption rates (OCR) in pmol $O_2$ per min. One day prior to the assay, organoids were re-seeded in 5 µL Matrigel-domes containing 15–20 intact organoids and covered with 200 µL of either WENR or ENR medium as indicated in XF96 cell culture microplates (Seahorse Bioscience; Cat: 102601-100). 1 h prior to the assay, the growth medium was discarded and replaced by Seahorse Assay medium (Seahorse Bioscience; Cat: 103059-000) and the plate was incubated for 60 min at 37 °C. For the mitochondrial stress test, culture medium was replaced by Seahorse XF Base medium (Seahorse Bioscience, Cat: 103575-100) supplemented with 20 mM glucose (Sigma-Aldrich), 2 mM L-glutamine (Sigma-Aldrich), 5 mM pyruvate (Sigma-Aldrich) and 0.56 ml NaOH (1 M). Mito Stress Test inhibitor stock solutions were prepared according to the kit manual (Seahorse Bioscience, Cat: 103015-100), adjusted to appropriate concentrations, and pipetted into the ports of the Sensor Cartridge (Port A: 20 µL of 10 µM Oligomycin; Port B: 22 µL of 5 µM FCCP; Port C: 25 µL of 2.5 µM Rotenone/ Antimycin A). After Seahorse measurement, the cell numbers in the wells were determined for normalization using the CyQuant NF Cell Quantification kit (Thermofisher, Cat: C35006). The obtained values were imported into the Wave Software and the normalized OCR- and ECAR-values were considered for analysis of the dataset.

**RNA extraction and qRT-PCR**. Total RNA was isolated from organoids using TRIreagent® (Thermofisher, Cat: 15596018), and cDNA was synthesized using the high-capacity cDNA reverse transcription kit (Fisherscientific, Cat: 10400745) according to manufacturer's instructions. Around 100–300 ng of total RNA was obtained from in vitro cultures. qPCR was performed with QuantStudio 6 Flex Real-time PCR System (Life technologies) using TAQMAN reagents and primers. CT values were normalized to *Hprt* using the ΔCT method. Taqman probes used in this paper are indicated below:

*Hprt* (Mm03024075_m1), *Ctgf* (Mm01192933_g1), *Ankrd1* (Mm00496512_m1), *Cyr61* (Mm00487498_m1), *Lgr5* (Mm00438890_m1), *Sox9* (Mm00448840_m1), *Ascl2* (Mm01268891_g1), *Slc26a3* (Mm00445313_m1), *Alpi* (Mm01285814_g1), *Car4* (Mm00483021_m1), *Clca3b* (Mm00519742_m1), *Muc2* (Mm01276696_m1), *Chga* (Mm00514341_m1), *Cdx2* (Mm01212280_m1)

**RNA sequencing**. WT and AHR-deficient generated from $n = 3$ biological replicates were cultured in WENR or d4 ENR (differentiation experiments) or stimulated with 5 nm FICZ (Sigma, Cat:SML1489) for 4 h at 37 C. Total RNA was isolated from organoids using TRIreagent®, and cDNA was synthesized using the high-capacity cDNA reverse transcription kit (Fisherscientific, Cat: 10400745) according to manufacturer's instructions. Total RNA was prepared into Illumina compatible libraries using the TruSeq Stranded RNA-Seq with RiboZero Gold (Illumina, Cat: 20020598) according to the manufacturer's instructions. Sequencing was carried out on the Illumina platform. Bam files were aligned TxDb.Mmusculus.UCSC.mm10.knownGene using GenomicAlignments summarizeOverlaps to produce raw gene counts. Further analysis was performed using DESeq2. PCA of the variance stabilized transform showed 90% of variance in the first PC corresponding to stem cells vs. differentiated organoids, 6% in the second PC corresponding to genotype and very low variance between replicates. Differentially expressed genes were called using a cutoff for adjusted *p* value of 0.05. Heatmaps using RNA-seq data were generated using Morpheus (https://software.broadinstitute.org/morpheus).

*Identification of activated transcriptional regulator and pathway analysis*. Predicted upstream transcription factors of DEGs were identified by using the IPA Core Analysis-Upstream Regulator tool (QIAGEN Inc., https://www.qiagenbioinformatics.com/products/ingenuity-pathway-analysis) where the analysis was restricted to identified upstream "transcriptional regulators". All identified predicted upstream regulators are reported in source data for Fig. 2D but only regulators with p-value of <0.0001 were considered as significant. Geneset enrichment analysis (GSEA) was conducted using the Broad institute GSEA tool (software.broadinstitute.org/gsea/index.jsp) using standard settings. We assessed overlaps with the H (hallmark gene sets), C5 (GO Biological processes geneset: Digestion, Intestinal Absorption) and C6 (Oncogenic signature genesets: Cordenonsi_Yap_conserved signature). For correlation with published data, expression dataset (.gct) files containing gene lists of DEGs identified in fetal spheroids or adult organoids were acquired from Mustata et al., 2013[24] the list of genes used to identity intestinal v gastric specific genes (selected from GTex repository) was acquired from Luknonin et al., 2020[20]. The CDX2KO organoid dataset (GSE62784) and the intestinal stem cell specific signature was taken from a single cell RNA-sequencing dataset, were originally from Simmini et al., 2014[47] and Haber et al, 2017[36] respectively.

**Omni-ATAC sequencing**. Wild type and AHR knockout colon organoids cultured in WENR or d4 ENR conditions were washed and digested with TrypLE buffer (Gibco, Cat: 12604013) to make single cell suspensions and Omni-ATAC was performed as described in Buenrostro JD et al[74]. In brief, 50,000 live cells were sort purified and immediately resuspended in 50 µl of ATAC-Resuspension Buffer (0.1% NP40, 0.1% Tween-20, and 0.01% Digitonin) and incubated on ice for 3 min, and nuclei was spin pelleted and transposition was carried out for 30 minutes at 37 C in a thermomixer. DNA was purified using the DNA Clean and Concentrator (Zymo Research, Cat: D4014) and transposed fragments were amplified for 10 cycles using NEBNext 2x MasterMix (NEB, Cat: M0541S) and adaptors to prepare libraries. Libraries were cleaned and quantified using Bioanalyzer followed by sequencing.

*Data analysis of ATAC-seq*. The nf-core/atacseq pipeline (version 1.2.1)[75] written in the Nextflow domain specific language (version 19.10.0)[76] was used to perform the primary analysis of the fastq samples in conjunction with Singularity (version 2.6.0)[77]. The command used was "nextflow run nf-core/atacseq -profile crick --input /Path_to_desing/design.csv --genome GRCm38". To summarise, the pipeline performs adapter trimming (Trim-Galore!. https://www.bioinformatics.babraham.ac.uk/projects/trim_galore/), read alignment (BWA) and filtering (SAMtools[78], BEDTools;[79] BamTools;[79] pysam - https://github.com/pysam-developers/pysam; picard-tools - http://broadinstitute.github.io/picard), normalised coverage track generation (BEDTools;[79] bedGraphToBigWig[80]), peak calling (MACS) and annotation relative to gene features (HOMER[81]), consensus peak set creation (BEDTools), differential binding analysis (featureCounts[82] R Core Team; DESeq2[83]) and extensive QC and version reporting (MultiQC;[75] FastQC;[84] deepTools;[85] ataqv[86]). All data was processed relative to the mouse UCSC mm10 genome. A set of consensus peaks was created by selecting peaks that appear in at least one sample. Counts per peak per sample were then imported on DESeq2 within R environment for differential expression analysis. Pairwise comparisons between genotypes in each condition, and between conditions per genotype were carried out and differential accessible peaks were selected with an FDR < 0.05. Heatmaps for differentially accessible peaks were generated using Deeptools.

For footprinting analysis TOBIAS (v 0.12.10)[32] was used by running the following pipeline (https://github.com/luslab/briscoe-nf-tobias). The pipeline runs TOBIAS' ATACorrect, ScoreBigwig, BINDetect and generates PlotAggregate metaplots on merged replicate bam files. TOBIAS was run on set of consensus peaks used for the differential analysis (see above), with the flag "--output-peaks" within TOBIAS BINDetect to set a different peak set for the output analysis. Differential binding scores and p-values are visualized as a volcano plot per pairwise comparison. JASPAR 2018 was used as the motif database for the foot printing analysis. As described before[32], all TFs with -log10(pvalue) above the 95% quantile or differential binding scores smaller/larger than the 5% and 95% quantile are colored. Selected TFs are also shown with labels. The full set of differential binding scores is included in the source data file. For integration of the AHR CHIP and ATAC data differential ChIP peaks and ATAC regions were obtained from DESeq (described in the ATAC and CHIP methods section in this document).

Transcription factor motifs were calculated from ATAC regions using TOBIAS (Bentsen et al., 2020). We subsetted the ATAC regions for evidence of AHR binding using bedtools by intersecting bed files made from the ChIP peaks and ATAC regions. We then scanned the ATAC regions also using bedtools intersect to find instances for each motif (Ahr, TEAD1, TEAD2, TEAD3 and TEAD4) where the whole motif was contained within the ATAC region. Motif co-occourance and distances were calculated using a custom jupyter notebooks python script. Additional python scripts exported the bed files with formatting appropriate for UCSC browser. All source code and analysis with instructions to reproduce the analysis can be found at: https://github.com/luslab/ahr-tead-motif-analysis

**Chromatin immunoprecipitation (ChIP) and sequencing.** WT and AHR knockout colon organoids generated from $n = 4$ biological replicates grown in WENR condition were stimulated with 5 nM FICZ for 1 hour in a glass vial at 37 °C. Samples were crosslinked with 1% formaldehyde (Thermofisher, Cat: 28908) for 8 min and quenched with 0.125 M glycine at room temperature. Nuclei was extracted by Dounce homogenization and chromatin was sheared by sonication in a lysis buffer (0.25% SDS, 1 mM EDTA, 1% Trition-X, 10 mM Tris-HCl, pH 8). Debris was spin pelleted at 13,000 rpm for 30 min at 4 °C. Supernatant was diluted in RIPA buffer (0.2% deoxycholate, 280 mM NaCl, 1.1% Triton-X in $H_2O$) and cleaned with protein A/G Dynabeads (Invitrogen, Cat: 10003D & 10001D) for 1 hour at 4 °C. 10% of supernatant was used as input and the remaining sample was immunoprecipitated overnight at 4 °C with anti-AhR antibody (Enzo, Cat: BML-SA210-0100). Chromatin was then incubated with protein A/G Dynabeads (Thermofisher, Cat: 88802) for 3 h at 4 °C. Chromatin bound beads were washed twice in low-salt buffer (0.1% SDS, 1% Triton X-100, 1 mM EDTA,, 140 mM NaCl, 0.1% deoxycholate, 20 mM Tris–HCl, pH 8), high-salt buffer (same as low salt except for 500 mM NaCl), LiCl wash buffer (250 mM LiCl, 0.5% NP40, 0.5% deoxycholate, 1 mM EDTA, 10 mM Tris–HCl, pH 8) and with Tris-EDTA buffer (10 mM Tris–HCl 1 mM EDTA). Protein-chromatin complexes were then eluted in elution buffer (1% SDS, 100 mM NaHCO3) and reverse cross-linked overnight with RNase and proteinase containing buffer (62.5 µg/ml RNaseA, 5 mg/ml proteinase K, 1.25 M NaCl, 62.5 mM EDTA pH 8, 250 mM Tris-HCl pH 6.5). ChIP DNA was purified using the ChIP DNA Clean and Concentrator (Zymo Research, Cat: D5205) and quantified using Bioanalyzer followed by sequencing.

*Data analysis for CHIP-seq.* Samples were sequenced on an Illumina HiSeq4000 generating 101 bp single ended reads averaging 30 million reads per sample. ChIP-seq reads were aligned to the mouse mm10 genome assembly using BWA version 0.7.15 [78]with a maximum mismatch of two bases. Picard tools version 2.1.1 (http://broadinstitute.github.io/picard) was used to sort, mark duplicates and index the resulting alignment BAM files. Normalised tdf files for visualisation purposes were created using the resulting BAM files using IGVtools (http://www.broadinstitute.org/igv) version 2.3.75 software by extending reads by 50 bp and normalising to 10 million mapped reads per sample. Peaks were called by comparing IP samples to their respective input using MACS version 2.1.1[87], using the standard parameters. Peaks called by MACS were annotated using the 'annotatePeaks' function in the Homer (version 4.8)[81] software package. Common and unique peaks across experiments were determined using the 'intersect' function in Bedtools (version 2.26.0)[79] and custom scripts.

**Reporting summary.** Further information on research design is available in the Nature Research Reporting Summary linked to this article.

## Data availability

Sequencing data associated with this paper has been submitted to NCBI's GEO repository under the accession number GSE179482. RNA-sequencing data of organoids grown in WENR and d4 ENR conditions can be found under accession number GSE133092. Source data are provided with this paper.

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

## Acknowledgements

This work was supported by the Francis Crick Institute, which receives its core funding from Cancer Research UK, The UK Medical Research Council, and the Wellcome Trust (FC001159). It was further supported by Wellcome Trust Grant 210556/Z/18/Z to B. Stockinger. F.G. was supported by an ERASMUS fellowship. For the purpose of Open Access, the author has applied a CC BY public copyright licence to any Author Accepted Manuscript version arising from this submission.' We would like to acknowledge the Science technology platforms at the Francis Crick Institute. We thank the Biological Research Facility for breeding and maintenance of our mouse strains, the Flow Cytometry Facility, the Advanced Sequencing Facility, the Light Microscopy Facility and the Histopathology Facility. We are grateful to Erik Sahai for critical comments on the manuscript.

## Author contributions

K.S. designed, performed and analysed most experiments. K.S. and B.S. wrote the manuscript. K.S. and M.M established the ATACseq datasets. M.M. established the ChIP datasets. M.L. and P.C. helped with the integration of ATACseq and ChIPseq datasets, F.G. carried out the metabolic experiments in Fig. 5., A.M. generated the RNAseq dataset, M.L. performed bioinformatic analysis on the ATACseq data, P.C. carried out bioinformatic analysis on the ChIPseq data, Y.L. assisted with histology experiments, M.T. helped with mouse DSS experiments, M.S. and G.K. conducted bioinformatic analysis of the RNA sequencing data, S.B. and D.B. purified and provided the Cyp1a1 inhibitor, C.C J.D.V. and M.L. performed TOBIAS analysis and B.S. supervised the project.

## Funding

## Competing interests
The authors declare no competing interests.
