## [Peer Review File · Nature Communications]

Cell-intrinsic Aryl Hydrocarbon Receptor signalling is required for the resolution of injury-induced colonic stem cellsEditorial Note: Parts of this Peer Review File have been redacted as indicated to remove third-party material where no permission to publish could be obtained.

Reviewers' Comments:

Reviewer #1:

Remarks to the Author:

Nuclear receptor AHR is an environmental sensor integrating microbial and dietary cues to modulate physiological and cellular processes within the intestinal epithelium. Understanding the role of AHR in intestinal homeostasis and regeneration is important. This study shows that AHR orchestrates the termination of the regenerative state through direct transcriptional regulation of genes involved in intestinal epithelial differentiation, and repression of genes involved in stem cell and wound-repair associated programs. An interesting finding is that mechanical sensing is linked to tissue regeneration via AHR.

The data are of high quality and compelling. They add to our understanding of the role of AHR specifically in intestinal epithelial cells where little is known (as opposed to other cell types that support intestinal functions). The data have implications for our understanding of the role of AHR in cancer processes and in the resolution of inflammatory processes.

I have several remarks that may lead to an improved manuscript:

As AHR is activated by various metabolic products, dietary cues and bacterially derived molecules, a key question remains how does the role of AHR in controlling regeneration link to AHR-activators, such as bacterial cues or other components? This has not been well addressed in this study, especially in the in vivo (mouse) model. Do activators of AHR modulate the regenerative time course in the mouse? Does stimulation of AHR by activators or bacterial components lead to a faster resolution of DSS wounds in the animal? How does this play in the organoid culture systems?

The AHR ChIP-seq was done in the presence of (AHR activator) FICZ but ATAC-seq, RNA-seq not (or did I miss something here?). So how does this all fit? Is there some component that weakly stimulates AHR in the organoid growth medium?

Other issues:

1.

What is the outcome in terms of disease index, weight loss etc during DSS and after of AHR deletion?

2.

Given the importance of the Yap1/ Tead complex, it may be good to know how Tead factors are expressed under the different conditions, e.g. AHR KO vs WT in ENR colon organoids (the authors state they did not find a difference in expression of Yap1 protein, this would be good to show as suppl. data, these days there should not be anymore 'data not shown').

Can the author perform a Tead factor Chip at select loci or Tead factor CHIP-seq?

3.

In the AHR ChIP-seq analysis, the authors identified 121695 peaks bound by AHR annotated to 18416 unique gene targets. What percentage of genes/ protein coding does this represent? How meaningful are these data, as they then narrow down the list by combining with the ATAC-seq-identified accessible sites (2785 unique genes)? What are the remaining presumed AHR binding sites?

4.

What is the relationship between AHR binding sites and Tead-motifs? Are these found clustered (on select genes) and if so what is the average distance between such sites? The data could be 'milked' more in this regard.

5.

When the authors perform an overlap analysis between DEG of RNA-seq and ATAC-seq (Fig 4B), it may be good to perform some Chi-square test or similar to show and give a value to significance; eyeing it, I suspect the overlap would be highly significant.

6.

Why use small intestinal scRNA-seq data set when colon data set is available (Tabula muris, Nature 2018 Oct;562(7727):367-372. doi: 10.1038/s41586-018-0590-4.)?

7.

Page 14, line 332, not clear: "We found that 13.52% of AHR targets overlap with approximately 21.42% of differentially upregulated and 19.63% of downregulated genes (Fig 6A, Supplemental Table 6)."

8.

Method, Primary culture of mouse colon organoids: what is D-PBS (page 22, line 526)?

9.

Legend Figure 1: why does (D) come before (C)? This does not make sense...

10.

Can the results of Fig. 1E be somehow quantified (especially for the KRT2 staining), as it is easy to pick some suggestive images...

11.

Legend, Figure 5F: "Metabolic analysis by Seahorse of AHR KO organoids compared with WT controls for basal OxPHOS vs glycolysis (left panel) and glycolytic index (right panel) and assessed at either 0h (WENR conditions), 48h (d2 ENR) and 96h (d4 ENR) post-Wnt removal" why are 4 bars shown/ panel, when only 3 time points analysed according to the legend?

12.

Fig 7 D, E: Can the changes in CDX2 and Sox9 staining be quantified?

13.

I do not understand the point of Fig.S3, showing ATAC-seq peaks across the entire genomes, especially as the libraries do not seem to be scaled to each other.

14.

The ATAC-seq bioinformatic analysis should be described in some more detail, e.g. how were MACS peaks defined and identified, what was the reference etc. Overall, the description of the bioinformatic analysis is a bit too terse for my taste.

15.

I do not quite understand this sentence, page 8, line 169 onwards: In d4 ENR conditions, Yap1, along with other proto-oncogenes genes such as Myc, β -catenin and Foxm1, were predicted to be activated upstream of genes upregulated in AHR KO compared to WT organoids (Fig. 2C)." What does the 'upstream' mean here?

Reviewer #2:

Remarks to the Author:

This is a very nice paper showing convincingly that, while the aryl hydrocarbon receptor (AHR) doesn't have much influence on intestine at steady state, it is important for resolution of the regenerative response and re-establishment of epithelial cell identity post insult. The authors use mice that express Villin-Cre and a floxed allele of AHR – a nice clean genetic system - to examine both intestinal physiology and use organoids derived from the mice for genomic interrogation. The conversion from fetal-like state back to committed epithelial cells is known to involve the ECM and the mechano-sensing transcription factors YAP and TAZ.

Specifically, what the authors show here is that loss of AHR causes failure to terminate the regenerative response post DSS-induced injury, including demonstrating persistent Sca-1 expression, and failure to re-differentiate efficiently. They used organoid cultures from the mice to perform RNAseq, CHIPseq and ATACseq, the analysis of these datasets showing that AHR deficiency alters chromatin accessibility, influencing YAP/TEAD transcriptional targets. AHR acts as transcriptional repressor for Sox9 and c-Myc (associated with stemness and proliferation), and an activator of Cdx2 (associated with differentiation) – providing likely plausible mechanisms for how AHR contributes to stopping regeneration and promotes differentiation after injury. As AHR is crucial in this, and since it responds to dietary and microbial-derived ligands, is it clearly at a pivotal place in maintaining tissue equilibrium, protecting against exaggerated inflammation and potentially cancer in the gut where

injury is happening all the time, via Wnt signalling. I think this work is important.

The data presented are generally clear and convincing. I am very supportive of publishing this work in Nature communications.

I have a few suggestions the authors might consider that could improve clarity of the main messages.

Specific suggestions:

1. Was there any change to organoid morphology as a result of AHR knock out?
2. Did the authors do the obvious experiment of examining nuclear levels of, for example, YAP – or has this been shown before to be regulated by AHR? It would be good confirmation of this part of their mechanistic hypothesis if feasible since it is standard read-out of pathway activation.
3. In Figure 3B, I was very interested in regulation of IL33. Could the authors comment of whether they know if this is nuclear (or secreted) IL33 and if it may be contributing to the transcriptional pathway – since it can bind to, and operate downstream of, FAK/integrin signalling and so may be part of the mechano-transduction effects.
4. The collagen experiment (changing the ratio of collagen / Matrigel, Fig 3F) is, for sure, consistent with a mechano-sensing influence on the genes shown; however, there may be other explanations for this, like different ratios of specific integrin heterodimer signalling etc. A more definitive experiment might be to block cellular tension with, for example, cytochalasin D (actin polymerisation inhibitor – a bit of a sledgehammer) or Y-27632 (ROCK inhibitor – a bit more subtle), if these agents are able to get into the cells of the organoids.
5. The difference in preference for glycolysis in AHR-deficient organoids when prompted to differentiate is interesting and consistent with hypotheses of the authors that there is impaired differentiation. It would be interesting to know if this feature is influenced by mechano-transduction/stiffness?
6. AHR KO and Cdx2 KO transcriptional parallels are compelling and suggest AHR is working via Cdx2 (is that correct?) If so, would a good experiment to test this directly be to enforce expression of Cdx2 in AHR null organoids and examine transcriptional changes, nuclear YAP levels etc, although I appreciate it is not a trivial experiment?
7. As sort of alluded to by point 5 above, I am a bit unsure who is regulating who in this 'transcriptional pathway' that clearly regulates the incredibly important biological response. I wonder if a model would help to delineate the respective relationships (upstream or downstream) of AHR, Cdx2, YAP/TEAD and downstream targets like Sox9, c-Myc in this pathway. I think it might help the reader.

Minor points:

8. Could the authors check acronyms are defined first time they are used, for example AOM on page 4.
9. On acronyms – I found the names of the conditions coupled with genotypes etc confusing and interrupted 'the nice read' of this manuscript – had to keep going back..... For example, d4 ENR AHR KO. I kept forgetting exactly what the annotations WENR and d4 ENR meant. Not sure if there is a simpler nomenclature to indicate proliferation or differentiation conditions. Or, the authors might consider putting their schematic in Fig S2 A into main figures when the model system introduced.
10. I think there is a stray reference not formatted - (Yan J et al, 2019, Gastroenterology) around page 20.

Margaret Frame, University of Edinburgh

Reviewer #3:

Remarks to the Author:

In the manuscript "Cell intrinsic Aryl Hydrocarbon Receptor signaling is required for the resolution of injury-induced colonic stem cells" by Shah et al., the authors carry out a series of experiments using AHR loss and gain of function models to identify the mechanism by which the Aryl Hydrocarbon receptor plays a role in resolution of the regenerative response after injury.

Overall, this is a very compelling manuscript with convincing data to support the main conclusions that AHR helps to control the transition from a regenerative state to a differentiated state by regulating chromatin accessibility. Specifically, the authors provide a model where AHR positively regulates Cdx2 expression, which in turn is required for intestinal epithelial differentiation.

The approaches used in this manuscript, including unbiased sequencing based approaches (RNA, ATAC, ChIP) lead to very convincing conclusions. The ability of the authors to compare transcriptional changes caused by AHR-LOF to CDX2-LOF, demonstrating that genes lost in CDX2-LOF are also lost in AHR-LOF and those gained in CDX2-LOF are gained in AHR-LOF combined with AHR ChIP-seq showing AHR to the CDX2 genomic locus is very compelling.

Overall, this is a very complete and mechanistic story that adds a lot to our understanding of how intestinal injuries are resolved. I only have a couple of minor concerns:

1. The data that AHR plays a role in resolution of the regeneration state, and subsequent acquisition of a terminally differentiated cell state, combined with data that loss of AHR leads to an enhanced stem cell/regenerative state is very well presented. Given this data, it would be good for the authors to show localization of AHR within the epithelium. Is its expression restricted to any particular compartment, or, is it ubiquitously expressed and its localization within the cell (ie. Nuclear vs. cytoplasmic) changes upon injury?
2. CDX2 immunostaining in figure 7D could be improved. Currently, there is quite a bit of background in the mesenchyme.
3. Given that AHR-LOF does not have a steady-state phenotype, it will be of interest for the authors to include in the discussion a point on how they think CDX2 may function without AHR in the healthy gut. It would be interesting if the authors comment on how they think AHR is getting activated during regeneration, and/or to speculate on a model for how AHR becomes engaged only during injury-repair. For example, is there a specific AHR ligand that comes into play during regeneration?

We would like to thank all reviewers for their thorough assessment of our manuscript and their constructive comments and will in the following answer the points that were raised.

Point-by Point response

Reviewer #1 (Remarks to the Author): *Nuclear receptor AHR is an environmental sensor integrating microbial and dietary cues to modulate physiological and cellular processes within the intestinal epithelium. Understanding the role of AHR in intestinal homeostasis and regeneration is important. This study shows that AHR orchestrates the termination of the regenerative state through direct transcriptional regulation of genes involved in intestinal epithelial differentiation, and repression of genes involved in stem cell and wound-repair associated programs. An interesting finding is that mechanical sensing is linked to tissue regeneration via AHR. The data are of high quality and compelling. They add to our understanding of the role of AHR specifically in intestinal epithelial cells where little is known (as opposed to other cell types that support intestinal functions). The data have implications for our understanding of the role of AHR in cancer processes and in the resolution of inflammatory processes. I have several remarks that may lead to an improved manuscript: As AHR is activated by various metabolic products, dietary cues and bacterially derived molecules, a key question remains how does the role of AHR in controlling regeneration link to AHR-activators, such as bacterial cues or other components? This has not been well addressed in this study, especially in the in vivo (mouse) model. Do activators of AHR modulate the regenerative time course in the mouse? Does stimulation of AHR by activators or bacterial components lead to a faster resolution of DSS wounds in the animal? How does this play in the organoid culture systems? The AHR ChIP-seq was done in the presence of (AHR activator) FICZ but ATAC-seq, RNA-seq not (or did I miss something here?). So how does this all fit? Is there some component that weakly stimulates AHR in the organoid growth medium?*

The reviewer queries the requirement for AHR ligands in the process of regulating regeneration. The functional activity of AHR is dependent on ligands, but in order to illustrate the role of ligands more directly, we included data on regeneration from DSS treatment in a previously described mouse strain (Schiering et al. Nature 542 (2017) with dysregulated Cyp1 expression, resulting in ligand depletion and as a consequence defective AHR activation. The data included in Fig.1E show that Villin-Cre $R26^{Cyp1a1}$ mice in which Cyp1a1 is constitutively expressed in intestinal epithelial cells, leading to rapid metabolic clearance of AHR ligands, phenocopy AHR deficient mice with similar increased levels of Sca-1 and failure to terminate the regenerative program. The defect in AHR signaling in these mice can be rescued by inhibition of the overactive Cyp1a1 enzyme. This is shown in Supplemental FigS1G where oral administration of a potent Cyp1a1 inhibitor (Sharma et al 2018) at different time points during the DSS phase shows that Cyp1a1 inhibition in the early regeneration phase will lead to reduction of Sca-1 expression and improved epithelial architecture on day 30 following DSS application.

Regarding the query about AHR ligands in the RNAseq vs ChIPseq approach, most culture media (including the one we are using for organoids) contain AHR ligands due to the obligatory presence of tryptophan. As shown by Wincent et al. (Ref.70) all commercial tryptophan is contaminated with small amounts of its metabolite FICZ. The reason for using FICZ stimulation prior to ChIPseq was simply to synchronise AHR transition to the nucleus. There are no AHR antibodies of high quality (hence the inclusion of an AHR knockout control) and it would simply be impossible to precipitate sufficient AHR if relying solely on the background FICZ levels.

Other issues: 1. What is the outcome in terms of disease index, weight loss etc during DSS and after of AHR deletion?

There are no weight differences observed in WT and IEC-AHR KO mice at steady-state. Both strains lose weight during the ulceration phase of the DSS-kinetic but regain weight during the repair phase. We have added these data to Fig.S1E.

2. *Given the importance of the Yap1/ Tead complex, it may be good to know how Tead factors are expressed under the different conditions, e.g. AHR KO vs WT in ENR colon organoids (the authors state they did not find a difference in expression of Yap1 protein, this would be good to show as suppl. data, these days there should not be anymore 'data not shown')*

We have added expression data for Teads under WENR and ENR conditions together with the Western blot showing similar expression of Yap protein in WT and AHR KO organoids (Supplementary Fig.S2C,D) .

Can the author perform a Tead factor Chip at select loci or Tead factor CHIP-seq?

A Tead factor ChIPseq would be beyond the scope of this manuscript. However, we have included an example showing Yap/Tead binding sites in a regulatory element upstream of the Sox9 promoter region which also contains several AHR binding sites in close proximity (Supplemental Fig.S5D).

3. *In the AHR CHIP-seq analysis, the authors identified 121695 peaks bound by AHR annotated to 18416 unique gene targets. What percentage of genes/ protein coding does this represent? How meaningful are these data, as they then narrow down the list by combining with the ATAC-seq-identified accessible sites (2785 unique genes)? What are the remaining presumed AHR binding sites?*

These targets will include not only what organoids are primed to express (in an AHR-dependent manner) upon differentiation, but also other targets that respond to AHR irrespective of the cellular state given the strong nature of the stimulation. We were also surprised at the large number of AHR-bound regions but given the absence of these targets in AHR KO control organoids we are confident that these represent real peaks. Therefore, to reduce the target list to what would be physiologically relevant targets of AHR in an organoid growing under WENR conditions, we restricted our analysis to AHR targets that were found to be accessible in colon organoids under WENR conditions. We have now included a pie chart with a % breakdown for the type of AHR targets in open chromatin regions in Supplemental Fig.S5B.

4. *What is the relationship between AHR binding sites and Tead-motifs? Are these found clustered (on select genes) and if so what is the average distance between such sites? The data could be 'milked' more in this regard.*

We have now conducted an analysis where we integrated the AHR CHIP and ATAC/TOBIAS data (D4 ENR AHR KO vs WT) focused on assessing the potential co-localization of TEAD and AHR in differentially accessible regions. This data can be found in a new supplemental Fig S6, showing the distance between AHR and TEAD motifs, classification of the regulatory elements these motifs can be found on, a GSEA (GO Biological process) to identify the biological function of nearby target genes as well as examples of key targets (UCSC tracks for Sox9, Itga2 and Nrp2). Further description of these findings are highlighted in the main text.

5. *When the authors perform an overlap analysis between DEG of RNA-seq and ATAC-seq (Fig 4B), it may be good to perform some Chi-square test or similar to show and give a value to significance; eyeing it, I suspect the overlap would be highly significant.*

This analysis has now been carried out- showing a high degree of significance- and the statistics are included in Fig.4B.

6. *Why use small intestinal scRNA-seq data set when colon data set is available (Tabula muris, Nature 2018 Oct;562(7727):367-372. doi: 10.1038/s41586-018-0590-4.)?*

We thank the reviewer for pointing out this paper as we were not previously aware of this important piece of work. However, in the Tabula Muris paper, only ~3-4k epithelial cells were sequenced from the large intestine compared to the ~53k epithelial cells sequenced in the Haber et al. paper. The annotations were also a lot broader than the Haber et al. paper. Specifically, there are no signatures for specialized epithelia such as Tuft, enteroendocrine and TA cells. Therefore, while we are grateful to the reviewer for directing us to this important piece of work, we feel that the Haber et al. paper is a better reference dataset for our work.

7. Page 14, line 332, not clear: “We found that 13.52% of AHR targets overlap with approximately 21.42% of differentially upregulated and 19.63% of downregulated genes (Fig 6A, Supplemental Table 6).”

We apologize for the lack of clarity, and have rephrased this sentence to make it clearer. We are referring to the percentage of AHR targets that overlap with the RNA-seq dataset.

8. Method, Primary culture of mouse colon organoids: what is D-PBS (page 22, line 526)?
D-PBS is just referring to the PBS (Ca⁺ and Mg⁺ free) that we use in the lab from GIBCO (Catalog number: 14190144)

9. Legend Figure 1: why does (D) come before (C)? This does not make sense...
We have modified Fig.1 to include additional data so this has now been resolved.

10. Can the results of Fig. 1E be somehow quantified (especially for the KRT2 staining), as it is easy to pick some suggestive images...

We have now included quantification for the images in Fig.1

11. Legend, Figure 5F: “Metabolic analysis by Seahorse of AHR KO organoids compared with WT controls for basal OxPHOS vs glycolysis (left panel) and glycolytic index (right panel) and assessed at either 0h (WENR conditions), 48h (d2 ENR) and 96h (d4 ENR) post-Wnt removal” why are 4 bars shown/ panel, when only 3 time points analysed according to the legend?

This was a mistake. The additional bar represented a 24h timepoint which we removed to maintain consistency with other datasets that focused on 0h, 48h and 96h.

12. Fig 7 D, E: Can the changes in CDX2 and Sox9 staining be quantified?

Quantifications for these images have been added to Fig.7

13. I do not understand the point of Fig.S3, showing ATAC-seq peaks across the entire genomes, especially as the libraries do not seem to be scaled to each other.

The libraries are scaled, but the peaks do not look uniform between the two conditions because organoids undergoing differentiation (WENR to d4 ENR) decrease chromatin accessibility in general. We felt it was important to show this general difference in chromatin accessibility between the stem cell and differentiated state.

14. The ATAC-seq bioinformatic analysis should be described in some more detail, e.g. how were MACS peaks defined and identified, what was the reference etc. Overall, the description of the bioinformatic analysis is a bit too terse for my taste.

We have extended the description of the bioinformatic analysis.

15. I do not quite understand this sentence, page 8, line 169 onwards: *In d4 ENR conditions, Yap1, along with other proto-oncogenes genes such as Myc, b-catenin and Foxm1, were predicted to be activated upstream of genes upregulated in AHR KO compared to WT organoids (Fig. 2C).*” What does the ‘upstream’ mean here?

These are transcription factors predicted by IPA (upstream regulator analysis) to have altered activity - either inhibited or activated - based on the transcriptional changes observed from the d4 ENR RNA-seq data we imported into the software. We have edited the text to clarify what we this term means.

Reviewer #2 (Remarks to the Author): *I have a few suggestions the authors might consider that could improve clarity of the main messages.*

Specific suggestions:

1. *Was there any change to organoid morphology as a result of AHR knock out?*

There were no changes in the morphology of organoids as a result of AHR deficiency.

2. *Did the authors do the obvious experiment of examining nuclear levels of, for example, YAP – or has this been shown before to be regulated by AHR? It would be good confirmation of this part of their mechanistic hypothesis if feasible since it is standard read-out of pathway activation.*

We agree with the reviewer’s point but did not do this analysis as YAP localization is variable in the whole population so it would better be addressed on the single cell level.

3. *In Figure 3B, I was very interested in regulation of IL33. Could the authors comment of whether they know if this is nuclear (or secreted) IL33 and if it may be contributing to the transcriptional pathway – since it can bind to, and operate downstream of, FAK/integrin signalling and so may be part of the mechano-transduction effects.*

This is a very interesting point worth exploring at some point, but we feel it is beyond the scope and focus of this current manuscript.

4. *The collagen experiment (changing the ratio of collagen / Matrigel, Fig 3F) is, for sure, consistent with a mechano-sensing influence on the genes shown; however, there may be other explanations for this, like different ratios of specific integrin heterodimer signalling etc. A more definitive experiment might be to block cellular tension with, for example, cytochalasin D (actin polymerisation inhibitor – a bit of a sledgehammer) or Y-27632 (ROCK inhibitor – a bit more subtle), if these agents are able to get into the cells of the organoids.*

We performed the suggested experiment and as the reviewer suggested Cytochalasin was too strong to reveal differences between the genotypes whereas using a Rock inhibitor we could clearly see that Yap target expression was strongly reduced in wildtype organoids but not in AHR deficient organoids albeit at high doses this difference was obscured. This indicates that indeed AHR deficient epithelial cells are hyperactive in terms of Yap signaling and not fully responsive to inhibition of the upstream pathway. We have included this in Fig.3G.

5. *The difference in preference for glycolysis in AHR-deficient organoids when prompted to differentiate is interesting and consistent with hypotheses of the authors that there is impaired differentiation. It would be interesting to know if this feature is influenced by mechano-transduction/stiffness?*

We agree with the reviewer that this would be interesting to explore but feel the experimental approach to tackle this with organoids would face several technical challenges (such as adherence of collagen to the seahorse plates) that would likely affect the readout.

6. AHR KO and Cdx2 KO transcriptional parallels are compelling and suggest AHR is working via Cdx2 (is that correct?) If so, would a good experiment to test this directly be to enforce expression of Cdx2 in AHR null organoids and examine transcriptional changes, nuclear YAP levels etc, although I appreciate it is not a trivial experiment?

The experiment suggested by the reviewer, while in principle feasible, might not provide conclusive results. While Cdx2 is important for intestinal cell fate determination, its dysregulation/over-expression has also been associated with certain malignancies. As we would have to add a Cdx2-transgene, we cannot be certain that the consequence of this artificial transduction would be representative of its restored physiological function. Furthermore, whether AHR's regulation of Cdx2 is linked to its regulation of Yap/Tead accessibility is not clear.

7. As sort of alluded to by point 5 above, I am a bit unsure who is regulating who in this 'transcriptional pathway' that clearly regulates the incredibly important biological response. I wonder if a model would help to delineate the respective relationships (upstream or downstream) of AHR, Cdx2, YAP/TEAD and downstream targets like Sox9, c-Myc in this pathway. I think it might help the reader.

We hope the reviewer will be satisfied that we will work on a cartoon once the paper is accepted.

Minor points: 8. Could the authors check acronyms are defined first time they are used, for example AOM on page 4.

We have added the definition for AOM= Azoxymethane in the text.

9. On acronyms – I found the names of the conditions coupled with genotypes etc confusing and interrupted 'the nice read' of this manuscript – had to keep going back..... For example, d4 ENR AHR KO. I kept forgetting exactly what the annotations WENR and d4 ENR meant. Not sure if there is a simpler nomenclature to indicate proliferation or differentiation conditions. Or, the authors might consider putting their schematic in Fig S2 A into main figures when the model system introduced.

We have taken the reviewer's advice and moved the schematic into the main Fig.2A to make the nomenclature clearer. One could potentially call these states stem cell vs differentiated but we feel it would be misleading as AHR KO do not really differentiate, whereas the WENR and ENR terms indicate the culture conditions that underlie these states.

10. I think there is a stray reference not formatted - (Yan J et al, 2019, Gastroenterology) around page 20.

We have corrected the formatting of this reference.

Reviewer #3 (Remarks to the Author):

I only have a couple of minor concerns: 1. The data that AHR is plays a role in resolution of the regeneration state, and subsequent acquisition of a terminally differentiated cell state, combined

with data that loss of AHR leads to an enhanced stem cell/regenerative state is very well presented. Given this data, it would be good for the authors to show localization of AHR within the epithelium. Is its expression restricted to any particular compartment, or, is it ubiquitously expressed and its localization within the cell (ie. Nuclear vs. cytoplasmic) changes upon injury?

We have shown in a previous publication (Schiering et al. Nature 2017) that AHR activation is visible throughout the epithelium and not confined just to stem cells. Here we include an image for the reviewer showing FACS plots of colon epithelial cells of an AHR reporter mouse (AHR tagged with a fluorescent reporter (AHR-Tomato)). It is difficult to comment on nuclear localisation as AHR has been shown to shuttle to the nucleus without intentional ligand activation (with the caveat that all culture media express AHR ligands). In our reporter mouse the fluorescent reporter is connected by a self-cleavable linker so that we cannot rely on presence of the fluorochrome in the nucleus alone.

[redacted]

i) Gating strategy for identification of live Epcam+ colonic epithelial cell types (Habowski et al. Comm. Bio 3 (2020) and Wang et al. Gastroenterology 145(2013) (ii) The dotted line depicts the MFI from a WT sample, the colors of the histogram plots and bar graphs correspond to epithelial sub-types.

2. CDX2 immunostaining in figure 7D could be improved. Currently, there is quite a bit of background in the mesenchyme.

We added quantification for Cdx2 in Fig.7. The apparent background is due to the fact that sometimes the staining is part of the crypt, not the mesenchyme.

3. Given that AHR-LOF does not have a steady-state phenotype, it will be of interest for the authors to include in the discussion a point on how they think CDX2 may function without AHR in the healthy gut. It would be interesting if the authors comment on how they think AHR is getting activated during regeneration, and/or to speculate on a model for how AHR becomes engaged only during injury-repair. For example, is there a specific AHR ligand that comes into play during regeneration?

AHR ligands in the body are not normally limiting since tryptophan, a precursor of many types of ligands is an essential amino acid. Furthermore, indoles derived from dietary intake provide ligands as does the metabolism of tryptophan by some microbiota species. We have included additional data in Fig.1E that emphasise the ligand dependency of AHR function in the regeneration process. Thus, it is unlikely and unnecessary to invoke a specific AHR ligand that might come into play during regeneration.

We hope that the reviewers are satisfied with our revisions and will now agree that this manuscript can be published.

Yours sincerely

Reviewers' Comments:

Reviewer #1:

Remarks to the Author:

The authors addressed all referees' comments well and robustly and improved (mostly by clarifying) what was already a very interesting manuscript containing high quality data. I am looking forward to see this important piece in press.

Patrick D. Varga-Weisz

Reviewer #2:

Remarks to the Author:

I am satisfied with all of the authors' responses to my comments. Overall, I think the manuscript is improved and is suitable for publication. It is interesting work.

M Frame

Reviewer #3:

Remarks to the Author:

It is a little disappointing that the authors chose not to address some very straight forward critiques, given the overall enthusiasm from the three reviewers.

In general, the revised manuscript is fine. As was pointed out initially, the quality of the immunostaining data in particular (Figure 7D and 7E) is not much improved. The authors did add quantification, which is not helpful if the quality of the data being quantified is low. For example, SOX9 expression in 7E is mostly cytoplasmic, which is unexpected for a transcription factor. Was cytoplasmic staining quantified?

With that said, I still feel this is a solid manuscript that is worthy of publication in Nat. Comm.

We would like to thank all reviewers for their positive comments on our revised manuscript.

Point-by Point response

Reviewer #3 (Remarks to the Author):

It is a little disappointing that the authors chose not to address some very straight forward critiques, given the overall enthusiasm from the three reviewers.

In general, the revised manuscript is fine. As was pointed out initially, the quality of the immunostaining data in particular (Figure 7D and 7E) is not much improved. The authors did add quantification, which is not helpful if the quality of the data being quantified is low. For example, SOX9 expression in 7E is mostly cytoplasmic, which is unexpected for a transcription factor. Was cytoplasmic staining quantified?

With that said, I still feel this is a solid manuscript that is worthy of publication in Nat. Comm.

The staining was done on OCT sections which were fixed 4% PFA for 2-3 hours, followed by an incubation in solution containing sucrose and glycerol. This was necessary as the Sca-1 staining does not work on standard paraffin sections. Since the identification of Sca-1⁺ epithelial cells was essential for the identification of regenerating foci, we also stained for Cdx2 and Sox9 in these sections. Using this method, we found that Sox9 was largely cytoplasmic in healthy adjacent tissue and stronger staining along with additional nuclear localization was present in regenerating foci and quantification was done for the whole crypt, including cytoplasmic and nuclear staining. Other publications (see links below) list similar observations for changing localization of Sox9 staining in different contexts.

Demarez C et al. (2016)

<https://journals.plos.org/plosone/article?id=10.1371/journal.pone.0157140#sec002>

Menzel-Severing J et al. (2018)

<https://www.nature.com/articles/s41598-018-28596-3#:~:text=Sox9%20localizes%20to%20the%20cytoplasm,during%20LEPC%20proliferation%20and%20differentiation>

Sumita Y et al. (2018)

<https://www.ncbi.nlm.nih.gov/labs/pmc/articles/PMC6151877/>

Yours sincerely